# Learning Infinitesimal Generators of Continuous Symmetries from Data

**Gyeonghoon Ko, Hyunsu Kim, Juho Lee**
Kim Jaechul Graduate School of AI
KAIST
Seoul, South Korea
`{kog, kim.hyunsu, juholee}@kaist.ac.kr`

## Abstract

Exploiting symmetry inherent in data can significantly improve the sample efficiency of a learning procedure and the generalization of learned models. When data clearly reveals underlying symmetry, leveraging this symmetry can naturally inform the design of model architectures or learning strategies. Yet, in numerous real-world scenarios, identifying the specific symmetry within a given data distribution often proves ambiguous. To tackle this, some existing works learn symmetry in a data-driven manner, parameterizing and learning expected symmetry through data. However, these methods often rely on explicit knowledge, such as pre-defined Lie groups, which are typically restricted to linear or affine transformations. In this paper, we propose a novel symmetry learning algorithm based on transformations defined with one-parameter groups, continuously parameterized transformations flowing along the directions of vector fields called infinitesimal generators. Our method is built upon minimal inductive biases, encompassing not only commonly utilized symmetries rooted in Lie groups but also extending to symmetries derived from nonlinear generators. To learn these symmetries, we introduce a notion of a validity score that examine whether the transformed data is still valid for the given task. The validity score is designed to be fully differentiable and easily computable, enabling effective searches for transformations that achieve symmetries innate to the data. We apply our method mainly in two domains: image data and partial differential equations, and demonstrate its advantages. Our codes are available at https://github.com/kogyeonghoon/learning-symmetry-from-scratch.git.

## 1 Introduction

Symmetry is fundamental in many scientific disciplines, crucial for understanding the structure and dynamics of physical systems, datasets, and mathematical models. The ability to uncover and leverage symmetries has become increasingly important in machine learning and scientific research due to its potential to improve model efficiency, generalization, and interpretability. By capturing inherent symmetrical properties, models can learn more compact and informative representations, leading to improved performance in tasks like supervised learning [31, 29, 4, 7, 33], self-supervised learning [8, 15, 23], and generative models [19, 11, 18].

Previous methods for learning symmetry have often relied on the explicit parameterization of group representations based on predefined generators, which can be limited in capturing various symmetries, including transformations that do not align along the generators. For example, when searching for Lie group symmetries in images or physics data, existing methods [3, 34] parameterize a group action $g$ as the matrix exponential of a linear combination of linear or affine Lie algebra generators $L_i$ with their learnable coefficients $w_i$ as $g = \exp\left(\sum_i w_i L_i\right)$. In the affine transformations of images in

38th Conference on Neural Information Processing Systems (NeurIPS 2024).

$(x_1, x_2)$-coordinates, there are six generators, each corresponding to translation, scaling, and shearing operations with respect to the $x_1$-axis and $x_2$-axis. Although there exist some methods that directly learn the generators, they are either bound to the general linear group $GL(n)$, which cannot account for non-affine or non-linear transformations [24], or are not guaranteed to find the correct symmetry in real-world image datasets [9, 13].

When searching for symmetries in high-dimensional real-world datasets, we can take advantage of the fact that the data can be interpreted as a function $f : \mathcal{X} \to \mathcal{Y}$, such as images, which are functions from the 2D Euclidean space to the color space. Another notable example of such data is partial differential equations (PDEs), where the data take the form $\boldsymbol{u} : \mathcal{X} \to \mathcal{U}$ and the Lie symmetries are defined as transformations on the space $\mathcal{X} \times \mathcal{U}$. There have been significant advances in Lie symmetry analysis in recent years, for both academic and industrial purposes, mostly involving extensive symbolic calculations and relying on computer algebra systems [26]. Discovering Lie symmetries of PDEs from data without prior knowledge is an unexplored topic, except for the work of Gabel et al. [14], which learns the symmetry generators of various PDEs in a supervised learning setup.

In this work, we propose a novel method for learning continuous symmetries, including non-affine transformations, from data without prior knowledge. By modeling one-parameter groups using Neural Ordinary Differential Equation (Neural ODE) [6], we establish a learnable infinitesimal generator capable of producing a sequence of transformed data through ODE integration. We design an appropriate *validity score* function that measures how much the transformation violates the invariance to certain criteria defined depending on the target task, and learn the generators by optimizing towards the validity score of the data transformed through ODE integration. For example, in an image classification dataset, we use a pre-trained feature extractor and define the validity score to be the cosine similarity between the features extracted from the original image and the transformed image. For PDEs, the validity score is defined by the numerical errors of the original equations after the transform. The validity scores are chosen based on the characteristics of the target tasks, and designed to be fully differentiable, so that the symmetry can be learned via gradient descent in an end-to-end fashion. We also incorporate two regularizations, orthonormality and Lipschitz loss, which prevent the learned generators from converging to a trivial solution.

Subsequently, we demonstrate that our method indeed discovers the correct symmetries in both image and PDE datasets. To the best of our knowledge, our research is the first to retrieve affine symmetry in the entire space of continuous transformations using the CIFAR-10 classification dataset, as shown in Figure 1. Moreover, our method excels in identifying non-affine symmetries and approximate symmetries in PDE tasks. We further demonstrate that the learned generators can be leveraged to develop automatic augmentation generators, which can be used to produce augmented training data for both image classification tasks and neural operator learning tasks of PDEs [21]. We provide empirical evidence that the models trained with data augmented by our learned generators perform competitively with those trained with traditional closed-form transforms such as Lie point symmetry (LPS) [4]. Moreover, we show that the *approximate symmetries* discovered by our method, which cannot be found by classical methods, can also boost the performance of the models, especially when the size of the training data is small.

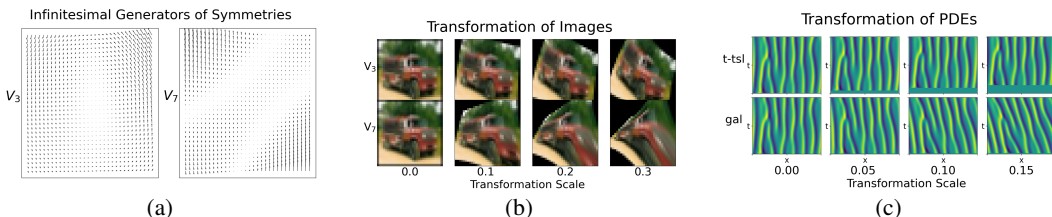

Figure 1: (a) Examples of the vector fields. $V_3$ is a learned symmetry which is approximately a rotation, while $V_7$ is not a symmetry, thus having a high validity score. (b) Transformed CIFAR-10 images using the learned generators. All the vector fields and transformations learned from CIFAR-10 are presented in Figure 8 of Appendix C. (c) Transformation of PDEs (KS equation) with learned symmetries: time translation (t-tsl) and Galilean boost (gal).

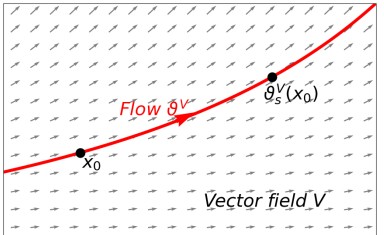

*Flow $\vartheta^V$*

$\vartheta_s^V(x_0)$

$x_0$

*Vector field V*

Figure 2: An example of a flow.

Table 1: Infinitesimal generators of the Affine group Aff(2). The set of six generators $\{L_1, \cdots, L_6\}$ forms a basis of the corresponding Lie algebra.

| Generator | Expression | One-Parameter Group | Description |
|---|---|---|---|
| $L_1$ | $(1, 0)$ | $\boldsymbol{x} \mapsto \boldsymbol{x} + s(1, 0)$ | translation in $x_1$-axis |
| $L_2$ | $(0, 1)$ | $\boldsymbol{x} \mapsto \boldsymbol{x} + s(0, 1)$ | translation in $x_2$-axis |
| $L_3$ | $(x_1, 0)$ | $\boldsymbol{x} \mapsto \left(\begin{smallmatrix} e^s & 0 \\ 0 & 1 \end{smallmatrix}\right)\boldsymbol{x}$ | scaling of $x_1$-axis |
| $L_4$ | $(0, x_2)$ | $\boldsymbol{x} \mapsto \left(\begin{smallmatrix} 1 & 0 \\ 0 & e^s \end{smallmatrix}\right)\boldsymbol{x}$ | scaling of $x_2$-axis |
| $L_5$ | $(x_2, 0)$ | $\boldsymbol{x} \mapsto \left(\begin{smallmatrix} 1 & s \\ 0 & 1 \end{smallmatrix}\right)\boldsymbol{x}$ | shear parallel to $x_1$-axis |
| $L_6$ | $(0, x_1)$ | $\boldsymbol{x} \mapsto \left(\begin{smallmatrix} 1 & 0 \\ s & 1 \end{smallmatrix}\right)\boldsymbol{x}$ | shear parallel to $x_2$-axis |
| $L_3 + L_4$ | $(x_1, x_2)$ | $\boldsymbol{x} \mapsto e^s\boldsymbol{x}$ | uniform scaling |
| $L_6 - L_5$ | $(-x_2, x_1)$ | $\boldsymbol{x} \mapsto \left(\begin{smallmatrix} \cos s & -\sin s \\ \sin s & \cos s \end{smallmatrix}\right)\boldsymbol{x}$ | rotation |

## 2 Preliminaries: One-parameter Group

In this section, we present the basic definitions of a one-parameter group, which we use to parameterize the symmetric transformations learned from the data.

Consider an unknown Euclidean domain $\mathcal{Z} \subseteq \mathbb{R}^n$ and a smooth vector field $V : \mathcal{Z} \to \mathbb{R}^n$. A path $\gamma : I = (a, b) \subseteq \mathbb{R} \to \mathcal{Z}$ satisfying $\frac{d}{ds}\gamma(s) = V(\gamma(s))$ for all $s \in I$ is a curve that travels around the domain $\mathcal{Z}$ with a velocity given by the vector field $V$. Along the curve $\gamma$, a point $\boldsymbol{x}_0 = \gamma(a_0)$ can be transported to $\gamma(a_0 + s)$ by flowing along the vector field $V$ by time $s$. We define the flow $\vartheta_s^V$ by $\vartheta_s^V(\boldsymbol{x}_0) = \gamma(a_0 + s)$ of $V$ as in Figure 2. This flow is computed by solving an ODE

$$\frac{d}{ds}\vartheta_s^V(\boldsymbol{x}) = V(\boldsymbol{x}) \tag{1}$$

with initial condition $\vartheta_0^V(\boldsymbol{x}) = \boldsymbol{x}$ for all $\boldsymbol{x} \in \mathcal{Z}$ [20].

The flow $\vartheta^V$ is governed by an autonomous ODE, i.e., an ODE independent of the temporal variable $s$. Due to properties of autonomous ODEs, the flow $\vartheta^V$ exists uniquely and it is smooth in both variables $s$ and $\boldsymbol{x}$. Assuming a mild condition on $V$, such as $V$ extends to a compactly supported vector field $\tilde{V}$ on $\mathbb{R}^n$, the ODE does not terminate in $\mathbb{R}^n$ in finite time and hence $\vartheta_s^V$ is defined for all $s \in \mathbb{R}$. In that case, the flow satisfies a group equation

$$\vartheta_{s_1+s_2}^V(\boldsymbol{x}) = \vartheta_{s_1}^V \circ \vartheta_{s_2}^V(\boldsymbol{x}) \tag{2}$$

for all $s_1, s_2 \in \mathbb{R}$. It means that the flow can be regarded as a group action of $\mathbb{R}$ on $\mathbb{R}^n$, transforming elements of $\mathcal{Z} \subseteq \mathbb{R}^n$. For this reason, $\vartheta_s^V$ is also called a *one-parameter group*, and the vector field $V$ is called an *infinitesimal generator* of the one-parameter group.

On $\mathcal{Z} = \mathbb{R}^n$, a constant vector field $V(\boldsymbol{x}) = \boldsymbol{v} \in \mathbb{R}^n$ gives rise to a translation $\boldsymbol{x} \mapsto \boldsymbol{x} + s\boldsymbol{v}$. For a matrix $\boldsymbol{A} \in \mathbb{R}^{n \times n}$, a vector field $V(\boldsymbol{x}) = \boldsymbol{A}\boldsymbol{x}$ gives rise to an affine transformation $\boldsymbol{x} \mapsto \exp(s\boldsymbol{A})\boldsymbol{x}$, where $\exp$ is the matrix exponentiation. Multiple infinitesimal generators may span a vector space $\mathfrak{g}$, and if $\mathfrak{g}$ satisfies some algebraic condition (closure under the Lie bracket), then $\mathfrak{g}$ forms a *Lie algebra*. Composing the elements of one-parameter groups of elements in $\mathfrak{g}$ gives rise to a *Lie group* $G$. The correspondence between $G$ and $\mathfrak{g}$ is called Lie group-Lie algebra correspondence.

*Continuous symmetries* are commonly defined by a Lie group $G$, acting on some domain and keeping the transformed objects *invariant* with respect to some criterion. We model symmetries by specifying their infinitesimal generators whose composition of one-parameter groups comprises the symmetries of that domain.

Below, we describe two representative examples that will be discussed extensively in the remainder of the paper: images (interpreted as functions on 2D planes) and PDEs.

### 2.1 Images and Their Symmetries

Consider a rescaled image of the form $f : \mathcal{X} = [-1, 1]^2 \subseteq \mathbb{R}^2 \to \mathcal{Y} = [0, 1]^3 \in \mathbb{R}^3$. The affine transformations on $\mathbb{R}^2$ have the form $\boldsymbol{x} = (x_1, x_2) \mapsto \boldsymbol{A}\boldsymbol{x} + \boldsymbol{b}$ for a matrix $\boldsymbol{A} \in \mathbb{R}^{2 \times 2}$ and a vector $\boldsymbol{b} \in \mathbb{R}^2$. The Affine transformations form the 6-dimensional Affine group Aff(2), and it has a corresponding 6-dimensional Lie algebra having a basis $\{L_1, \ldots, L_6\}$ given as in Table 1. The symmetries of images are often exploited as a data augmentation strategy for learning image classifiers, under an assumption that the transforms do not alter the identity or semantics of the images to be classified.

## 2.2 PDEs and Their Symmetries

Given an $n$-dimensional independent variable $\boldsymbol{x} = (x_1, \cdots, x_n) \in \mathcal{X} \subseteq \mathbb{R}^n$ and an $m$-dimensional dependent variable $\boldsymbol{u} = \boldsymbol{u}(\boldsymbol{x}) = (u_1(\boldsymbol{x}), \cdots, u_m(\boldsymbol{x})) \in \mathcal{U} \subseteq \mathbb{R}^m$, we denote by $\boldsymbol{u}^{(i)}$ the collection of all $i$-th partial derivatives of $\boldsymbol{u}$ with respect to $\boldsymbol{x}$. A *partial differential equation* $\boldsymbol{\Delta}$ on $\boldsymbol{u}(\boldsymbol{x})$ of order $k$ is defined by a set of algebraic equations $\boldsymbol{\Delta}(\boldsymbol{x}, \boldsymbol{u}, \boldsymbol{u}^{(1)}, \cdots, \boldsymbol{u}^{(k)}) = 0$ involving all the variables and their partial derivatives. For example, two scalar independent variables $x, t \in \mathbb{R}$ and one scalar dependent variable $u(x,t) \in \mathbb{R}$ governed by equation $\Delta = u_t + uu_x + u_{xxx} = 0$ gives the 1-dimensional Korteweg-de Vries (KdV) equation, where we denote partials using subscripts, e.g. $u_t = \frac{\partial u}{\partial t}$ and $u_{xxx} = \frac{\partial^3 u}{\partial x^3}$. The KdV equation is commonly used to model the dynamics of solitons, e.g. shallow water waves [36]. The KdV equation described above is an example of 1-dimensional scalar-valued evolution equation. Such an equation takes the form $u = u(x,t) \in \mathbb{R}$ with its governing equation of the form

$$u_t = F(x, t, u, u_x, u_{xx}, u_{xxx}, \cdots) \tag{3}$$

for some function $F$. In this paper, we only deal with 1D scalar-valued evolution equation on a fixed periodic domain $x \in [0, L]$.

Continuous symmetries of PDEs are commonly parametrized by a one-parameter group on $\mathcal{X} \times \mathcal{U}$. Denote $(\boldsymbol{\xi}, \boldsymbol{\mu}) = (\boldsymbol{\xi}[\boldsymbol{x}, \boldsymbol{u}], \boldsymbol{\mu}[\boldsymbol{x}, \boldsymbol{u}])$ an infinitesimal generator defined on $\mathcal{X} \times \mathcal{U}$. Then the PDE $\boldsymbol{\Delta}$ possesses the infinitesimal generator of symmetry $(\boldsymbol{\xi}, \boldsymbol{\mu})$ if the equation is still satisfied after transforming both the independent variable $\boldsymbol{x}$ and the dependent variable $\boldsymbol{u}$ [27, 5, 26]. Symmetries of PDEs are categorized by how the generators $(\boldsymbol{\xi}, \boldsymbol{\mu})$ depend on $(\boldsymbol{x}, \boldsymbol{u})$. The symmetry is a Lie point symmetry (LPS) if the value of $(\boldsymbol{\xi}, \boldsymbol{\mu})$ at each point $(\boldsymbol{x}, \boldsymbol{u}(\boldsymbol{x}))$ depends only on the point value $(\boldsymbol{x}, \boldsymbol{u}(\boldsymbol{x}))$ itself. If $(\boldsymbol{\xi}, \boldsymbol{\mu})$ also depends on the derivatives $\boldsymbol{u}^{(1)}, \cdots, \boldsymbol{u}^{(k)}$ at that point, it is called a Lie-Bäcklund symmetry or generalized symmetry. If $(\boldsymbol{\xi}, \boldsymbol{\mu})$ depends on integrals of $\boldsymbol{u}$, then it is called a nonlocal symmetry. Finding an LPS of a PDE $\boldsymbol{\Delta}$ can be done algorithmically under some mild assumptions on $\boldsymbol{\Delta}$. However, there is no general recipe of finding Lie-Bäcklund symmetries or nonlocal symmetries, and discovering such symmetries remains an active area of research.

## 3 Related Work

**Symmetry discovery.** Approaches to learning symmetries can be categorized by addressing two questions: (a) *where do they search for symmetries*, and (b) *what are they aiming to learn*. One line of research aims to learn ranges, focusing on determining the ranges of transformation scales that enhance learning when employed as augmentation techniques. For example, Benton et al. [3] learns transformation ranges of predefined transformations by treating them as learnable parameters and backpropagating through differentiable transformations.

Another line of research aim to learn subgroups of bigger candidate groups, typically a linear group $GL(n)$ or an affine group $\text{Aff}(n)$. For example, Desai et al. [10] use the Generative Adversarial Network (GAN) to search for symmetries, with the generator transforming data by group elements sampled from the candidate group and the discriminator verifying whether the transformed data sill lies in the data distribution. Similarly, Yang et al. [34] employ the GAN approach, but generator of GAN models infinitesimal generators instead of the subgroup itself, and learns affine symmetries such as rotation of images and Lorentz symmetry of high-energy particles. As an alternative, Moskalev et al. [24] proposed an idea of extracting symmetries from learned neural network by differentiating through it, and retrieved 2D rotation in the linear group using the rotation MNIST dataset.

Finally, *learning symmetries with minimal assumption*, i.e. without assuming the infinitesimal generators are linear or affine, is an area of large interest. An early attempt of Rao & Ruderman [28] models infinitesimal generator by a learnable matrix from the pixel space to the pixel space, and learn 2D rotation by solving a task that compares original images and rotated ones, where the images are $5 \times 5$ random pixels. Sohl-Dickstein et al. [32] takes the similar approach with eigen-decomposing the learnable matrix. Dehmamy et al. [9] builds a convolution operation whose kernel encodes learnable infinitesimal generators, and retrieved 2D rotation from random $7 \times 7$ images by comparing original and transformed ones, and Yang et al. [35] uses an autoencoder to simplify nonlinear symmetries into linear ones. Our work closely aligns with Liu & Tegmark [22] and Forestano et al. [13], which model one-parameter groups by an MLP and learn the symmetries from an invariant scalar quantity. To the

best of our knowledge, learning correct symmetries with minimal assumption was only achieved with toy datasets, far from real-world datasets such as CIFAR-10.

**Utilizing symmetries in deep learning.** An effective method for leveraging symmetries in deep learning is data augmentation [31]. In the image domain, there are numerous augmentation techniques available [29], most of which are based on geometric properties of images. Although data augmentation techniques have been primarily explored in the context of images, recent studies by [4, 23] have demonstrated that symmetries can also be used for augmenting data in the training of neural PDE solvers. In addition to data augmentation, some approaches involve designing new neural network architectures that inherently reflect the group symmetries of the input data [7]. Wang et al. [33] applied a similar strategy within the PDE domain.

# 4 Learning Continuous Symmetries with One-Parameter Groups

## 4.1 Training Process

Given a learning task with a dataset $\mathcal{D} \subset \mathcal{A}$ in an underlying space $\mathcal{A} = \{f | f : \mathcal{X} \to \mathcal{Y}\}$, we aim to model symmetry by a one-parameter group $\vartheta_s$ acting on $\mathcal{A}$, as explained in § 4.3. We define a continuous symmetry by stating that $\vartheta$ is a symmetry of this task if there exists some $\sigma > 0$ such that for any data point $f \in \mathcal{D}$ and transformation scale $s \in [-\sigma, \sigma]$, the transformed data point $\vartheta_s(f)$ remains valid for this task. We assume the existence of a differentiable *validity score* $S(\vartheta_s, f) \in \mathbb{R}$, such that $\vartheta_s(f) \in \mathcal{A}$ is valid if $S(\vartheta_s, f) < C$ for a certain threshold $C \in \mathbb{R}$. Then, a one-parameter group $\vartheta$ is a symmetry of the task if $S(\vartheta_s, f) < C$ for all $f \in \mathcal{D}$.

The validity score depends on the nature of the target task, though no strict criterion exists. As long as it is differentiable and the valid data aids learning, it is considered acceptable. For instance, we can define the validity based on a negative log-likelihood of a probabilistic model. In § 4.2, we discuss the validity scores to be used for image and PDE data.

Once a validity score is defined, we learn a symmetry $\vartheta^*$ by minimizing the validity scores of transformed data,

Figure 3: Process of learning symmetry.

$$\vartheta^* = \arg\min_{\vartheta} \mathbb{E}_{f \sim \mathcal{D}, s \sim \mathrm{Unif}([-\sigma, \sigma])} \left[ S(\vartheta_s, f) \right], \quad (4)$$

where the $\arg\min$ is taken over the entire class of smooth one-parameter groups. Since the learning is performed in function space, we appropriately constrain the function space using a regularizer, as described in § 4.4. Once symmetries are learned, they reveal the symmetrical properties of the target task, which can then be exploited to augment the training data.

## 4.2 Task-specific Definition of Validity Score $S$

**Images.** In image-related tasks, we define a validity score using a pre-trained neural network. Let $\mathcal{D}$ be an image classification dataset consisting of data of the form $(f, y) \in \mathcal{A} \times \mathbb{R}$, where $f$ is an image and $y$ is a label. Also let $H_{\mathrm{cls}} \circ H_{\mathrm{fext}} : \mathcal{A} \to \mathbb{R}$ be a learned neural network, where we denote by $H_{\mathrm{fext}} : \mathcal{A} \to \mathbb{R}^k$ the feature extractor and $H_{\mathrm{cls}} : \mathbb{R}^k \to \mathbb{R}$ the classifier. We define the validity score $S(\vartheta_s, f)$ as the cosine similarity between the features before and after the transformation:

$$S(\vartheta_s, f) = \mathrm{sim}\left( H_{\mathrm{fext}}(\vartheta_s(f)), H_{\mathrm{fext}}(f) \right), \quad (5)$$

where $\mathrm{sim}$ is the cosine similarity defined as $\mathrm{sim}(\boldsymbol{v}_1, \boldsymbol{v}_2) = \frac{|\boldsymbol{v}_1 \cdot \boldsymbol{v}_2|}{\|\boldsymbol{v}_1\| \|\boldsymbol{v}_2\|}$ for all $\boldsymbol{v}_1, \boldsymbol{v}_2 \in \mathbb{R}^k \setminus \{\boldsymbol{0}\}$.

**PDEs.** Let $\boldsymbol{u}(\boldsymbol{x})$ be a solution of a given PDE $\boldsymbol{\Delta}$, discretized on a rectangular grid $\mathcal{X}_{\mathrm{grid}} = \{\boldsymbol{x}_i\}_{i=1}^{N_{\mathrm{grid}}}$. For a transformed data $\vartheta_s(\boldsymbol{u})$, we measure the violation of the equality $\boldsymbol{\Delta} = 0$ to assess whether the transformed data is still a valid solution. Using an appropriate numerical differentiation method, we directly compute the value of the PDE, denoted as $\boldsymbol{\Delta}(\vartheta_s(\boldsymbol{u}))$, which represents the error of $\vartheta_s(\boldsymbol{u})$

as a solution of $\mathbf{\Delta}$, taking a value $\mathbf{\Delta}(\vartheta_s(\boldsymbol{u}))_i$ at grid point $\boldsymbol{x}_i$. The validity score is defined by the summation of all PDE errors across the grid points:

$$S(\vartheta_s, \boldsymbol{u}) = \sum_i |\mathbf{\Delta}(\vartheta_s(\boldsymbol{u}))_i|. \tag{6}$$

For example, for a solution $u(x)$ of the 1D KdV equation, we examine whether the transformed solution $\tilde{u} = \vartheta_s(u)$ satisfies $\tilde{u}_t + \tilde{u}\tilde{u}_x + \tilde{u}_{xxx} = 0$ where the partials are computed using a numerical differentiation method.

## 4.3 Parametrization of One-Parameter Groups using Neural ODE

On a Euclidean domain $\mathcal{Z} \subseteq \mathbb{R}^n$, we model an infinitesimal generator with an MLP $\boldsymbol{h_\theta} : \mathcal{Z} \subseteq \mathbb{R}^n \to \mathbb{R}^n$. The infinitesimal generator $\boldsymbol{h_\theta}$ gives rise to a one-parameter group $\vartheta_s^{\boldsymbol{h_\theta}}$. We sample a transformation scale $\alpha \sim \text{Unif}([-\sigma, \sigma])$ for a predefined hyperparameter $\sigma \in \mathbb{R}_{>0}$. To transform a point $x \in \mathcal{Z}$ along this one-parameter group by an amount $\alpha \geq 0$, we use a numerical ODE solver to solve the ODE for $\gamma : [0, \alpha] \to \mathcal{Z}$ satisfying

$$\gamma'(s) = \boldsymbol{h_\theta}(\gamma(s)), \quad \forall s \in [0, \alpha], \quad \gamma(0) = x \tag{7}$$

and obtain a transformed data point $\tilde{\boldsymbol{x}} = \vartheta_\alpha^{\boldsymbol{h_\theta}}(\boldsymbol{x}) = \gamma(\alpha)$. If $\alpha < 0$, we compute $\vartheta_\alpha^{\boldsymbol{h_\theta}}(\boldsymbol{x}) = \vartheta_{-\alpha}^{-\boldsymbol{h_\theta}}(\boldsymbol{x})$ by integrating $-\boldsymbol{h_\theta}$ instead of $\boldsymbol{h_\theta}$ using the ODE solver. We can backpropagate through the numerical ODE solver using the adjoint method [6] to learn $\theta$.

Let $f : \mathcal{X} \to \mathcal{Y}$ be a data point on a domain $\mathcal{A}$. As $\mathcal{A}$ is a space of functions, naïvely modeling symmetry on $\mathcal{A}$ may ignore the geometry implied in the input space $\mathcal{X}$. Instead, we define two transformations: $\vartheta_\mathcal{X}$ on $\mathcal{X}$ and $\vartheta_\mathcal{Y}$ on $\mathcal{Y}$, and induce a transformation of $f$ by

$$(\vartheta_\mathcal{X}(f))(x) = f(\vartheta_\mathcal{X}^{-1}(x)), \quad (\vartheta_\mathcal{Y}(f))(x) = \vartheta_\mathcal{Y}(f(x)), \tag{8}$$

where we abuse notation and write the transformed function as $\vartheta_\mathcal{X}(f)$ and $\vartheta_\mathcal{Y}(f)$. For an image represented as a discretized function $f : \mathcal{X} \to \mathcal{Y}$ from $\mathcal{X} = [-1, 1]^2$ and $\mathcal{Y} = [0, 1]^3$, $\vartheta_\mathcal{X}$ corresponds to spatial transformations such as translation or rotation, and $\vartheta_\mathcal{Y}$ corresponds to color space transformations. For a PDE, a 1D scalar-valued evolution equation on a fixed periodic domain takes the form $u(x, t) \in \mathcal{U} = \mathbb{R}$ with $(x, t) \in [0, L] \times [0, T] = \mathcal{X} \subseteq \mathbb{R}^2$, and we parameterize an infinitesimal generator on a product space $\mathcal{X} \times \mathcal{U} \subseteq \mathbb{R}^3$ by an MLP. Then, a transformation on $(x, t, u) \in \mathcal{X} \times \mathcal{U}$ induces a transformation on the solution of the PDE $u(x, t)$.

## 4.4 Objective Functions

**Symmetry loss.** Let $N_{\text{sym}}$ be the number of symmetries to be learned. Let $(\boldsymbol{h_\theta}^{(a)})_{a=1}^{N_{\text{sym}}}$ be the infinitesimal generators computed from a single MLP. For each $a \in \{1, \ldots, N_{\text{sym}}\}$, we sample a transformation scale $s_a \sim \text{Unif}([-\sigma, \sigma])$ to transform $f$ via numerical integration. The parameter $\boldsymbol{\theta}$ is optimized by minimizing the average validity score over the training data,

$$\mathcal{L}_{\text{sym}}(\boldsymbol{\theta}) = \sum_{a=1}^{N_{\text{sym}}} \mathbb{E}_{f \sim \mathcal{D}, s_a \sim \text{Unif}([-\sigma, \sigma])} \left[ S\left( \vartheta_{s_a}^{\boldsymbol{h_\theta}^{(a)}}, f \right) \right]. \tag{9}$$

**Orthonormality loss.** Learning only with the symmetry loss may result in trivial solutions such as the zero vector field or the same vector field repeated in multiple slots. To prevent this, we introduce the orthonormality loss to regularize the model towards learning orthonomral vector fields. Specifically, given two vector fields $V_1, V_2 : \mathcal{Z} \to \mathbb{R}^n$, we define an inner product as,

$$\langle V_1, V_2 \rangle = \frac{1}{\text{vol}(\mathcal{Z})} \int_\mathcal{Z} \omega(\boldsymbol{x})(V_1(\boldsymbol{x}) \cdot V_2(\boldsymbol{x})) d\boldsymbol{x} \approx \frac{1}{|\mathcal{Z}_{\text{grid}}|} \sum_{\boldsymbol{x}_i \in \mathcal{Z}_{\text{grid}}} \omega(\boldsymbol{x}_i)(V_1(\boldsymbol{x}_i) \cdot V_2(\boldsymbol{x}_i)), \tag{10}$$

with a suitable weight function $\omega(x) : \mathcal{Z} \to \mathbb{R}$ and a discretized grid $\mathcal{Z}_{\text{grid}}$ of $\mathcal{Z}$ of size $|\mathcal{Z}_{\text{grid}}|$. Given this definition, we first normalize each generator by its norm to ensure $\|\boldsymbol{h_\theta}^{(a)}\|^2 = 1$. Then we compute the orthonormality loss as,

$$\mathcal{L}_{\text{ortho}}(\boldsymbol{\theta}) = \sum_{1 \leq a < b \leq N_{\text{sym}}} \left\langle \text{sg}(\boldsymbol{h_\theta}^{(a)}), \boldsymbol{h_\theta}^{(b)} \right\rangle, \tag{11}$$

where $\mathrm{sg}(\cdot)$ denotes the stop-gradient operation to ensure that the constraint $\langle \boldsymbol{h}_{\boldsymbol{\theta}}^{(a)}, \boldsymbol{h}_{\boldsymbol{\theta}}^{(b)} \rangle = 0$ only affects the latter slot ($b$). By doing this, if the true number of symmetries $N_{\mathrm{sym}}^*$ is less than or equal to the assumed number of symmetries $N_{\mathrm{sym}}$, the learned symmetries will be aligned in the first $N_{\mathrm{sym}}^*$ slots.

**Lipschitz loss.** We further introduce inductive biases to the infinitesimal generators we aim to learn. For instance, an infinitesimal generator moving only a single pixel near the boundary by a large scale would be undesirable. This idea can be implemented using Lipschitz continuity. For a grid point $\boldsymbol{x}_i \in \mathcal{Z}_{\mathrm{grid}}$ and its neighboring point $\boldsymbol{x}_j \in \mathrm{nbhd}(\boldsymbol{x}_i) \subset \mathcal{Z}_{\mathrm{grid}}$, we expect the vector field $V$ to satisfy the Lipschitz condition,

$$\mathrm{Lips}(V; \boldsymbol{x}_i, \boldsymbol{x}_j) < \tau \text{ where } \mathrm{Lips}(V; \boldsymbol{x}_i, \boldsymbol{x}_j) = \frac{\|V(\boldsymbol{x}_i) - V(\boldsymbol{x}_j)\|}{\|\boldsymbol{x}_i - \boldsymbol{x}_j\|}. \tag{12}$$

To regularize the model toward the Lipschitz condition, we introduce the Lipshictz loss,

$$\mathcal{L}_{\mathrm{Lips}}(\boldsymbol{\theta}) = \sum_{a=1}^{N_{\mathrm{sym}}} \sum_{\boldsymbol{x}_i \in \mathcal{Z}_{\mathrm{grid}}, \boldsymbol{x}_j \in \mathrm{nbhd}(\boldsymbol{x}_i)} \max(\mathrm{Lips}(\boldsymbol{h}_{\boldsymbol{\theta}}^{(a)}; \boldsymbol{x}_i, \boldsymbol{x}_j) - \tau, 0). \tag{13}$$

**Total loss and loss-scale-independent learning.** We jointly minimize the three loss functions with suitable weights $w_{\mathrm{sym}}, w_{\mathrm{ortho}}, w_{\mathrm{Lips}} > 0$ and learn the weights $\boldsymbol{\theta}$ of MLP using a stochastic gradient descent:

$$\boldsymbol{\theta}^* = \arg\min_{\boldsymbol{\theta}} w_{\mathrm{sym}} \mathcal{L}_{\mathrm{sym}}(\boldsymbol{\theta}) + w_{\mathrm{ortho}} \mathcal{L}_{\mathrm{ortho}}(\boldsymbol{\theta}) + w_{\mathrm{Lips}} \mathcal{L}_{\mathrm{Lips}}(\boldsymbol{\theta}). \tag{14}$$

To minimize the computational burden of hyperparameter tuning, we ensure that all the loss terms have a *natural scale*, i.e. a dimensionless scale independent of the context. For example, when penalizing the inner product in Equation 11, we apply $\arccos$ to the normalized inner product to ensure the loss term lies in $[0, \pi/2]$. Similarly, the scale of the PDE validity score $S(\vartheta_s, \boldsymbol{u})$ in Equation 9 depends on the scale of the data $\boldsymbol{u}$. When penalizing it, we apply the log function so that the gradients are scaled automatically as $\nabla_{\boldsymbol{\theta}} \log(S(\vartheta_s, \boldsymbol{u})) = \nabla_{\boldsymbol{\theta}} S(\vartheta_s, \boldsymbol{u}) / S(\vartheta_s, \boldsymbol{u})$.

Here we describe the generic training process, but the actual implementation requires non-trivial task-specific designs, such as the choice of the weighting function $w(\boldsymbol{x})$ or the method for locating the transformed data on the target grid. We defer these details for image and PDE tasks to Appendix A.

### 4.5 Comparison With Other Methods

Here, we compare our method with other recent symmetry discovery methods. The differences mainly arise from (a) what they aim to learn (e.g., transformation scales or subgroups from a larger group) and (b) their assumptions about prior knowledge (e.g., complete, partial, or no knowledge of symmetry generators). Another important distinction is the viewpoint on symmetry: some methods learn symmetries that raw datasets inherently possess (implicit), while others learn symmetries from datasets explicitly designed to carry such symmetries (explicit).

Some recent symmetry discovery works are listed in Table 2. We emphasize that our method excels in two key aspects: (a) our learning method reduces infinitely many degrees of freedom, (b) our method works with high-dimensional real-world datasets. For example, while LieGAN [33] and LieGG [24] reduce a 6-dim space (affine) to a 3-dim space (translation and rotation) in an image dataset, ours reduces an $\infty$-dim space to a finite one. L-conv [9] also does not assume any prior knowledge, but it is limited in finding rotation in a toy task, where it learns rotation angles of rotated images by comparing them with the original ones, which are 7x7 random pixel images.

## 5 Experiments

### 5.1 Images

We use images of size $32 \times 32$ from the CIFAR-10 classification task. Since our method does not model discrete symmetry, we use horizontal flip with 50% probability by default. We train a ResNet-18 model, which will be used as the feature extractor $H_{\mathrm{fext}}$ in Equation 5. The weight function

Table 2: Comparison with other symmetry discovery methods.

| | **Augerino** | **LieGAN** | **LieGG** | **L-conv** | **Forestano et al.** | **Ours** |
|---|---|---|---|---|---|---|
| **Symmetry generators** | completely known | partially known (affine) | partially known (affine) | completely unknown | completely unknown | completely unknown |
| **Learn what?** | transformation scales | symmetry generator (rotation / Lorentz) | symmetry generator (rotation) | symmetry generator (rotation) | symmetry generator (in low-dim task) | symmetry generator (affine) |
| **Verified with what?** | raw CIFAR-10 | rotation MNIST / Top tagging | rotation MNIST | random $7 \times 7$ pixel image | toy data (dim $\leq$ 10) | raw CIFAR-10 & PDEs |
| **Implicit or explicit?** | implicit | explicit | explicit | explicit | explicit | implicit |
| **How?** | optimize while training downstream task | compare fake/true data in GAN framework | extracts from learned NN using Lie derivative | compare rotated and original images | extracts from invariant oracle using Lie derivative | extracts from validity score using ODE integration |

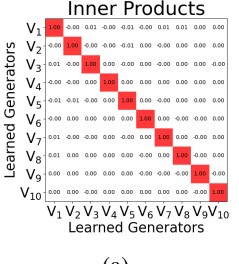

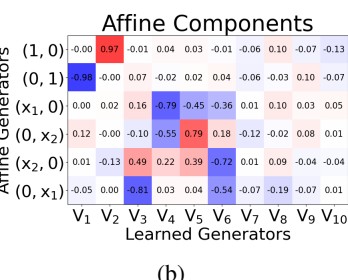

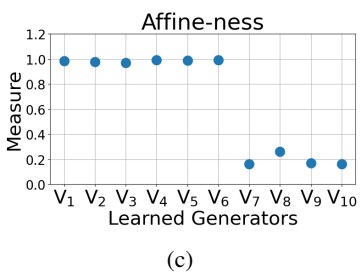

| (a) | (b) | (c) |
|---|---|---|

Figure 4: (a) Self inner-products of the learned generators. (b) Inner product comparison of the learned generators with the affine generators. (c) Affine-ness of learned generators.

on the pixels is computed as explained in Appendix A.1. We expect to find 6 affine generators and we use an MLP modeling 10 vector fields in the pixel space $[-1, 1]^2 \subseteq \mathbb{R}^2$, expecting the first six learned vector fields to be the affine generators. We learn the Equation 14 using stochastic gradient descent with $w_{\text{sym}} = 1$ and $w_{\text{ortho}}, w_{\text{Lips}} = 10$. The parameter $\sigma$, which controls the scale of transformation, is set to $\sigma = 0.4$, and the Lipschitz threshold $\tau$ is set to $\tau = 0.5$. Other details are described in Appendix B.1. We conducted three trials with random initializations and report the full results in Appendix C.1. Furthermore, we also learn symmetries in the *color space*, and their results are shown in Appendix G.

**Learned symmetries.** Since we expect to learn affine symmetries, we compare the results with the affine basis $\{L_1, \cdots, L_6\}$ defined in Table 1. We compute the inner products $\langle V, L_i \rangle$ of the learned vector field $V$ with $L_i$ for $i = 1, \cdots, 6$ to measure how much the learned vector fields contain the affine basis and measure the affine-ness of vector field by Affine-ness$(V)^2 = \sum_{i=1}^{6} \langle V, L_i \rangle^2$.

In all experiments, we successfully retrieve six linearly independent affine generators in the first six slots. Figure 4a shows that the learned generators are orthogonal to each other, as desired. Figure 4b shows the inner product between the learned generators and the affine generators. Since the affine-ness measure of the first 6 learned generators in Figure 4c is almost close to 1, we can read out the affine components in Figure 4b and say that e.g., $V_1 \approx (0, -0.98 + 0.12x_2)$. Notably, two translation generators are found in the first two slots, indicating that the two translations are *the most invariant* one-parameter group among the entire class of one-parameter groups on the pixel space. After the two translation generators, four affine generators are learned, indicating that affine transformations are *the next most invariant* transformations. In particular, the third and fourth generators are close to the rotation generator and the scaling generator, respectively. The remaining four generators fix pixels close to the center and transform boundary pixels by a large magnitude.

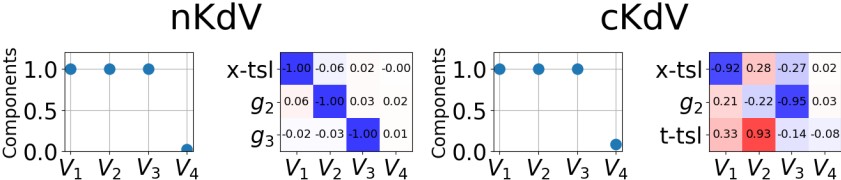

Figure 5: Inner products between the learned non-affine symmetry generators and the ground truth. The results including the affine symmetry generators are shown in Figure 10a.

Table 4: Definition of PDEs.

| Name | Equation |
|------|----------|
| KdV | $u_t + uu_x + u_{xxx} = 0$ |
| KS | $u_t + u_{xx} + u_{xxxx} + uu_x = 0$ |
| Burgers | $u_t + uu_x - \nu u_{xx} = 0$ |
| nKdV | $e^{-\frac{\hat{t}}{t_0}} u_{\hat{t}} + uu_x + u_{xxx} = 0$ |
| cKdV | $u_t + uu_x + u_{xxx} + \frac{u}{2(t+1)} = 0$ |

Table 5: LPS of PDEs.

| Name | Lie Point Symmetries | | |
|------|-----|-----|-----|
| KdV [4], KS [4], Burgers [17] | $(1,0,0)$ | $(0,1,0)$ | $(t,0,1)$ |
| nKdV (Appendix D.3) | $(1,0,0)$ | $(0, e^{-\frac{\hat{t}}{t_0}}, 0)$ | $(t_0(e^{\frac{\hat{t}}{t_0}} - 1), 0, 1)$ |
| cKdV [30] | $(1,0,0)$ | $(\sqrt{t+1}, \frac{1}{2\sqrt{t+1}}, 0)$ | |

**Analysis.** Unlike other symmetry discovery researches [24, 9] that use datasets which are explicitly designed to be symmetric such as rotation MNIST, we discovered affine transformations from CIFAR-10 with no augmentation except horizontal flip. Moreover, we extract the symmetries from the ResNet trained with CIFAR-10 without augmentation. This implies that although CIFAR-10 is not explicitly composed to be symmetric under affine transformations, the dataset possesses intrinsic affine symmetry. This also implies that even the ResNet trained without augmentation possesses invariance under affine transformation. It is widely believed that the strong generalization power of neural networks is linked to the augmentation insensitivity of the neural networks [25]. Our results show that ResNet is insensitive to affine transformation even when not explicitly designed to be so, supporting this hypothesis. This result implies that augmentation explicitly amplifies the insensitivity by using transformed data for training. Our analysis is tangential to that of Gruver et al. [16], which measures the extent of invariance using derivatives along the infinitesimal generators instead of ODE integration.

**Augmentation results.** We train a ResNet-18 model using CIFAR-10 classification data, applying the learned symmetries as data augmentation. We compare the results of no-augmentation, default augmentation (horizontal flip and random crop), and affine transformation, with transformation scale searched in $\{0.1, 0.2, 0.3, 0.4, 0.5\}$. We conduct five experiments with random initialization for all the settings and report the results in Table 3.

Table 3: Test accuracy in CIFAR-10.

| Method | Acc. (%) |
|--------|----------|
| No-aug | $92.4 \pm 0.3$ |
| Default | $\mathbf{95.1 \pm 0.1}$ |
| Affine | $\mathbf{95.1 \pm 0.2}$ |
| **Learned** | $94.9 \pm 0.1$ |

## 5.2 PDEs

We follow the experimental setting of Brandstetter et al. [4], which use the Korteweg-de Vries (KdV) equation, the Kuramoto-Shivashinsky (KS) equation, and the Burgers' equation on a 1D periodic domain as experiments. They all have time translation, space translation, and the Galilean boost as LPSs. To consider PDEs with non-trivial and non-affine symmetries, we add two variants of the KdV, namely the nKdV equation and the cKdV equation. The nKdV is yielded by a nonlinear time translation of the original KdV, and the cKdV is the cylindrical KdV equation, having an extra time-dependent term. The equations are listed in Table 4, and their symmetries are listed in Table 5. Note that some symmetries are ruled out since we work within a fixed periodic domain, e.g., the scaling symmetry of the KdV. Since there are at most three symmetries, we open four slots in an MLP and learn symmetries using weights $w_{\text{sym}}, w_{\text{Lips}} = 1$ and $w_{\text{ortho}} = 3$.

**Learned symmetries.** We compare the learned symmetries with the ground-truth symmetries using inner products. We found the ground truth symmetries in all the experiments as in Figure 5 and Figure 10.

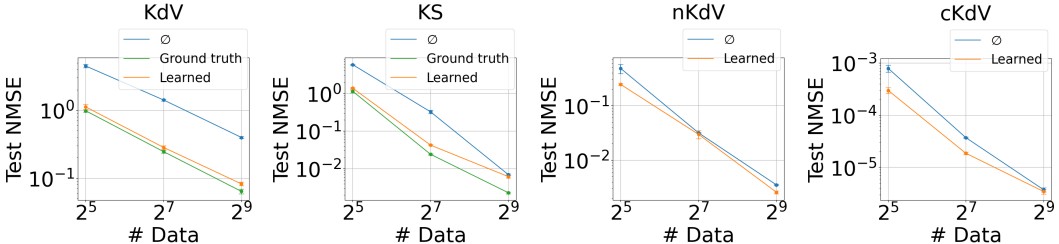

Figure 6: Comparison of augmentation performances using the ground truth symmetries and the learned symmetries with various numbers of data. The symbol $\emptyset$ stands for no-augmentation.

Table 6: Test NMSE comparison of augmentation using LPS and AS in FNO learning for cKdV.

| # Data | None | LPS | AS | LPS+AS |
|--------|------|-----|-----|--------|
| $2^7$ | $(3.70 \pm 0.09) \times 10^{-6}$ | $(3.49 \pm 1.06) \times 10^{-6}$ | $(3.20 \pm 0.94) \times 10^{-6}$ | $(2.66 \pm 0.36) \times 10^{-6}$ |
| $2^5$ | $(7.90 \pm 1.21) \times 10^{-4}$ | $(5.90 \pm 0.17) \times 10^{-4}$ | $(4.45 \pm 0.15) \times 10^{-4}$ | $(3.70 \pm 0.15) \times 10^{-4}$ |

Interestingly, for the Burgers' equation and the cKdV equation, we additionally found the $u$-axis rescaling operation $(0, 0, u)$ and the time translation $(0, 1, 0)$ respectively. Applying the $u$-axis rescaling $u \mapsto cu$ for $c \approx 1$ to the Burgers' equation $u_t + uu_x - \nu u_x x = 0$ gives $cu_t + c^2 uu_x - \nu cu_x x = c(c-1)uu_x$, leaving only the $uu_x$ term. The $uu_x$ term in Burgers' is called the *convection term*, and it is approximately zero in most region and spikes in some small region. Similarly, the time translation $t \mapsto t + c$ for $c \approx 0$ fixes the first three terms in the cKdV equation $u_t + uu_x + u_{xxx} + u/(2(t+1)) = 0$ and only changes the last term $u/(2(t+1))$ by a negligible amount. These are not LPSs of the given equations, but the error of the PDE after transformation is smaller than the error of the numerical differentiation method. These are *approximate symmetries* (AS), and the theory of AS is also of great interest in the symmetry analysis of PDEs [2, 1].

**Augmentation results.** We use the learned symmetries as data augmentation and train Fourier Neural Operators (FNOs). The detailed experiment setting is described in Appendix B.2. Since FNOs are extremely sensitive to numerical error, we employ Whittaker-Shannon interpolation, explained in Appendix D.2, to resample the transformed results. The results are depicted in Figure 6. In all cases, data augmentation using the learned symmetries improve the performance, almost close to the results using the ground truth symmetries. The detailed results are in Appendix C.2. Additionally, we verify that the approximate symmetry of cKdV is also beneficial to training, as shown in Table 6, especially when the numbers of data points is low, proving the effectiveness of symmetries extracted from data.

**Ablations.** Additional ablation studies on numerical methods, such as numerical differentiation and interpolation, and hyperparameter sensitivity are conducted, and their results are presented in Appendices E and F.

## 6 Conclusion

We have introduced a novel method for learning continuous symmetries, including non-affine transformations, from data without prior knowledge. By leveraging Neural ODE, our approach models one-parameter groups to generate a sequence of transformations, guided by a task-specific validity score function. This approach captures both affine and non-affine symmetries in image and PDE datasets, enhancing automatic data augmentation in image classification and neural operator learning for PDEs. The learned generators produce augmented training data that improve model performance, particularly with limited training data.

**Limitation.** However, despite its flexibility, our method requires careful selection of numerical methods, such as numerical differentiation and interpolation, to ensure stable training and the ODE integration can be computationally large for augmentation generation compared to other augmentation methods. While we focus on image symmetries and LPSs of PDEs, the method could potentially model other symmetries and domains with proper validity scores, suggesting future applications in learning complex symmetries, including conditional and non-local symmetries, in various data types.

## Acknowledgments and Disclosure of Funding

This work was partly supported by Institute of Information & communications Technology Planning & Evaluation (IITP) grant funded by the Korea government(MSIT) (No.RS-2019-II190075, Artificial Intelligence Graduate School Program(KAIST), No.2022-0-00713, Meta-learning Applicable to Real-world Problems, No.2022-0-00184, Development and Study of AI Technologies to Inexpensively Conform to Evolving Policy on Ethics, No.RS-2024-00509279, Global AI Frontier Lab)

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

# A Implementation Details

## A.1 Image Dataset

**Notation.** Images are functions of the form $f : \mathcal{X} \to \mathcal{Y}$, where $\mathcal{X} = [-1, 1]^2$ is the spatial domain and $\mathcal{Y} = [0, 1]^3$ is the normalized RGB domain. Pixels $\mathcal{X}_{\text{grid}} = \{\boldsymbol{x}_i\}_{i=1}^{N_{\text{grid}}}$ are discretized through a rectangular grid of $\mathcal{X}$, and images $f$ are discretized on the pixel space by $f_i = f(\boldsymbol{x}_i)$ for all $i$.

$\omega(\boldsymbol{x})$ **in orthonormality loss.** We first learn a symmetry on the spatial domain $\mathcal{X}$ using a feature extractor $H_{\text{hidden}}$ taken from a pretrained neural network. One obstacle is that in most image datasets, the main subjects of images are mostly located around the centers, and the regions close to the boundary are filled with backgrounds. In other words, each pixel has a different level of importance, and we may end up learning infinitesimal generators that only move boundary pixels and fix the center.

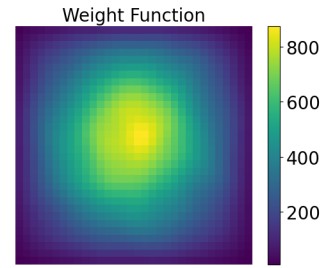

Weight Function

Figure 7: The weight function $\omega(\boldsymbol{x})$.

We take into account the importance of pixels using the weight function $\omega(x) : \mathcal{X} \to \mathbb{R}_{\geq 0}$. For a discretized image $f = \{f_i\}$ and for each pixel $\boldsymbol{x}_i$, we measure a *pixel sensitivity* of the image $f$ at the $i$-th pixel up to the feature extractor $H_{\text{fext}}$ as

$$\text{Sensitivity}(f, \boldsymbol{x}_i) = \left\| \frac{\partial H_{\text{fext}}(f)}{\partial \boldsymbol{x}_i} \right\| = \left\| \frac{\partial f_i}{\partial \boldsymbol{x}_i} \frac{\partial H_{\text{fext}}(f)}{\partial f_i} \right\|, \quad (15)$$

which can be computed by querying a Jacobian-vector product of $H_{\text{fext}}$ with respect to the image gradient at each pixel $\frac{\partial f_i}{\partial \boldsymbol{x}_i}$. We define the weight $\omega(x)$ as the average of pixel sensitivity across the dataset:

$$\omega(\boldsymbol{x}_i) = \mathbb{E}_{f \sim \mathcal{D}}[\text{Sensitivity}(f, \boldsymbol{x}_i)] = \mathbb{E}_{f \sim \mathcal{D}} \left[ \left\| \frac{\partial H_{\text{fext}}(f)}{\partial \boldsymbol{x}_i} \right\| \right]. \quad (16)$$

However, computing this weight function $\omega$ needs $N_{\text{grid}} \cdot N_{\text{data}}$ times of computation of Jacobian-vector products. Instead, we use a Gaussian kernel $\kappa(\boldsymbol{x}; \hat{\boldsymbol{x}}, \sigma) = \frac{1}{\sqrt{2\pi}\sigma} \exp(-\frac{\|\boldsymbol{x} - \hat{\boldsymbol{x}}\|^2}{2\sigma^2})$, with a fixed $\sigma = 0.1$ and the center $\hat{\boldsymbol{x}}$ sampled uniformly on $\mathcal{X}$, and approximate the weight function by

$$\omega(\boldsymbol{x}_i) \approx \mathbb{E}_{f \sim \mathcal{D}, \hat{\boldsymbol{x}} \sim \text{Unif}(\mathcal{X})} \left[ \kappa(\boldsymbol{x}_i; \hat{\boldsymbol{x}}, \sigma) \left\| \sum_j \frac{\partial H_{\text{fext}}(f)}{\partial \boldsymbol{x}_j} \kappa(\boldsymbol{x}_j; \hat{\boldsymbol{x}}, \sigma) \right\| \right] \quad (17)$$

so that we compute the weight in a stochastic manner. Intuitively, it computes Jacobian-vector product for each larger pseudo-pixel represented by the Gaussian kernels instead of each individual pixel. We iterate over the dataset 20 times, and the computed weight function is shown in Figure 7.

**Training details.** The infinitesimal generator $h_{\boldsymbol{\theta}}^{(a)}$ transforms each pixel $\boldsymbol{x}_i$ into $\tilde{\boldsymbol{x}}_i$ via ODE integration. In this process, the transformed pixels $\{\tilde{\boldsymbol{x}}_i\}$ may no longer be located on the rectangular grid $\mathcal{X}_{\text{grid}}$, so we use the bilinear interpolation method to resample the transformed image $\vartheta_s(f)$ on $\mathcal{X}_{\text{grid}}$. Note that the bilinear interpolation operation is differentiable, thereby we can train the MLP by minimizing the validity score $S(\vartheta_s, f)$, which is the cosine similarity between the features of $f$ and $\vartheta_s(f)$:

$$S(\vartheta_s, f) = \text{sim}(H_{\text{fext}}(\vartheta_s(f)), H_{\text{fext}}(f)). \quad (18)$$

## A.2 PDE Dataset

**Notation.** Solutions of a 1D scalar-valued evolution equation on a periodic domain take the form $u(x, t) : \mathcal{X} \to \mathcal{U}$ for $\mathcal{X} = [0, L] \times [0, T]$ and $\mathcal{U} = \mathbb{R}$, where $[0, L]$ is the spatial domain and $[0, T]$ is the temporal domain. Similar to the image task, the domain $\mathcal{X}$ is discretized by the rectangular grid $\mathcal{X}_{\text{grid}} = \{\boldsymbol{x}_i\}_{i=1}^{N_{\text{grid}}} = \{(x_i, t_i)\}_{i=1}^{N_{\text{grid}}}$. A discretized solution is a set of tuples $(x_i, t_i, u_i)$ where $u_i = u(x_i, t_i)$.

$\omega(x)$ **in orthonormality loss.** Unlike in images, we assume that all discretized grid points hold equal importance, so we set $\omega(x) = 1$ for all $x \in \mathcal{X}$.

**Training details.** The infinitesimal generators on the product space $\mathcal{X} \times \mathcal{U}$ transform a point $\boldsymbol{u} = \{(x_i, t_i, u_i)\}$ into $\tilde{\boldsymbol{u}} = \{(\tilde{x}_i, \tilde{t}_i, \tilde{u}_i)\}$. Therefore, the transformed solutions are no longer on the rectangular grid. To compute the PDE value $\Delta(\tilde{\boldsymbol{u}})$ for $\tilde{\boldsymbol{u}} = \vartheta_s(\boldsymbol{u})$ in Equation 6, we need to compute partial derivatives numerically such as $u_x(\tilde{x}_i, \tilde{t}_i)$, $u_t(\tilde{x}_i, \tilde{t}_i)$ or $u_{xx}(\tilde{x}_i, \tilde{t}_i)$. We use the *finite difference method*, in which the numerical derivative is approximated by finite differences, e.g.,

$$u_x(x_i, t_i) = \frac{u(x_i + \Delta x, t_i) - u(x_i - \Delta x, t_i)}{2\Delta x}. \tag{19}$$

for some small $\Delta x$.

In particular, we use the weighted essentially non-oscillating (WENO) scheme as a numerical differentiation method [37], with a careful choice of parameters as in Dumbser & Käser [12]. In the WENO method, multiple estimates for derivatives are made using multiple sets of neighboring gridpoints (called *stencils*). The multiple estimates are then averaged with weights (called *smoothness indicator*) that approximate how stable the derivative estimates are . We implement the WENO method working on a nonuniform grid, and the model learns symmetries by backpropagating through it. A detailed description on the WENO scheme is in Appendix D.1.

# B  Experiment Details

In this section, we present the detailed experimental settings of the experiments in § 5.

## B.1  Images

**Training ResNet-18.** When training the ResNet-18 with CIFAR-10, both the feature extractor $H_{\text{fext}}$ and models after augmentation, we train the model in 200 epochs with a batch size 128. The learning rate is set to $10^{-1}$ and decreases by a factor of 0.2 at the 60th, 120th, and 160th epoch. The model is trained by SGD optimizer with Nesterov momentum 0.9 and weight decay 0.4.

**Learning symmetries.** To learn the symmetry generators, we train the MLP using two shared hidden layers, each with a width of 256, followed by a hidden layer of width 32 for each output vector field. We use the swish activation function to ensure the learned vector fields are smooth. The MLP is trained for 50 epochs with a batch size 128 and fixed learning rate of $10^{-4}$ using the Adam optimizer. The learning process takes less than 10 hours on a GeForce RTX 2080 Ti GPU.

## B.2  PDEs

**Data generation.** We follow the data generation method of Brandstetter et al. [4]. Given an 1D evolution equation $u_t = F(x, t, u, u_x, u_xxx, \cdots)$ for $u = u(x, t)$ on periodic domain $[0, L]$, we start with an initial condition $u(x, 0)$ by random Fourier series as

$$u(x, 0) = \sum_{p=1}^{P} A_p \sin(2\pi l_p x / L + \phi_p) \tag{20}$$

where $P$ is the number of Fourier modes and $(A_p, l_p, \phi_p)$ are random coefficients. For time-stepping, we compute $x$-derivatives $u_x, u_{xx}, \cdots$ using pseudospectral method, which computes derivatives in Fourier domain and converts them back to the original domain. We use an ODE solver to compute the time evolution of $u(x, t)$ from $t = 0$ to $t = T$. The solution is discretized on regular grid of size $N_x \times N_t$ on $[0, L] \times [0, T]$, where $N_x = 256$ and $N_t = 140$ by default.

When simulating Burgers' equation, we instead solve the PDE for the heat equation $\phi_t = \phi_{xx}$ for $\phi = \nu\phi(x, t)$. The Burgers' equation $u(x, t)$ and the heat equation $\phi(x, t)$ is related via the Cole-Hopf transformation:

$$u = 2\nu \frac{\partial}{\partial x} \log(x). \tag{21}$$

After data generation, we transform the heat equation back to the Burgers' equation.

**Learning symmetries.** We use an MLP with the same architecture as described in Appendix B.1. Since all the tuples $(x, t, u)$ must pass through the MLP, totaling $N_x \times N_t$ for each data instance, we set a small batch size 4. Symmetries are learned using 1024 data instances over 50 epochs. We use the Adam optimizer and train the network with a learning rate of $10^{-4}$ in the first 25 epochs and $10^{-5}$ for the remaining 25 epochs. The learning process also takes less than 10 hours on a GeForce RTX 2080 Ti GPU.

**Sobolev regularization.** To ensure smoothness in the vector fields, we apply additional regularization using the Sobolev norm of order 2 in the $x$-domain. For a vector field $V(x, t)$, the Sobolev norm can be efficiently computed in Fourier domain:

$$\|V(\cdot, t)\|_{2,2}^2 = \sum_{i=0}^{2} \left\| \frac{\partial^i V(\cdot, t)}{\partial x^i} \right\|_2^2 = \sum_{n=0}^{N_x - 1} \left( 1 + |\frac{n_{\text{freq}}}{L}| \right)^2 \hat{V}(n, t) \tag{22}$$

where $n_{\text{freq}} = \min(n, N_x - n)$ and $\hat{V}(\cdot, t)$ is the discrete Fourier transformation of $V(\cdot, t)$. Since we already enforce $\|V\| = 1$, we penalize towards the Sobolev norm excluding the zeroth order term:

$$\|V(\cdot, t)\|_{2,2}^2 - \|V(\cdot, t)\|_2^2 = \sum_{i=1}^{2} \left\| \frac{\partial^i V(\cdot, t)}{\partial x^i} \right\|_2^2 = \sum_{n=0}^{N_x - 1} \left( \left( 1 + |\frac{n_{\text{freq}}}{L}| \right)^2 - 1 \right) \hat{V}(n, t). \tag{23}$$

We apply the Sobolev regularization during the final 10 epochs.

**Training FNOs.** For the KdV and KS equations, we train an autoregressive FNO solver, which takes 20 timesteps as input and predicts the subsequent 20 timesteps, following the experimental setup of Brandstetter et al. [4]. For the nKdV and cKdV equations, which are time-dependent and contain explicit $t$ terms, we train FNOs as single-time neural operators. These models utilizes the initial conditions of the equations to predict the states after 70 timesteps. We train the FNO over 40 epochs, with each epoch comprising 280 iterations across the dataset for the KdV and KS equation and 100 iterations for the nKdV and cKdV equations. The learning rate begins at $10^{-4}$ and decreases by a factor of $0.4$ every 10 epochs.

## C   Experiment Results

In this section, we provide detailed results of the experiments outlined in § 5. For symmetry learning tasks, we conducted each experiment three times and randomly selected one for reporting in § 5. The complete set of results is provided here. Also, we report the detailed evaluation metrics in augmentation tasks.

### C.1   Images

Figure 8 is a visualizations of the learned symmetries discussed in § 5.1. Figure 9 displays results from experiments conducted under the same settings but with different model initializations compared to Figure 4 in § 5.1. It is notable that affine symmetries consistently occupy the first six slots across all experiments.

### C.2   PDEs

We report experiments results for learning symmetries, conducting three trials for each equation in figure Figure 10. The ground truth symmetries and the approximate symmetries are consistently found in the former slots in all the experiments. The remaining fourth slots occasially converge to learned symmetries despite the orthonormality loss or converge to some unknown vector fields with high validity scores. The results from the first trials are used in the augmentation experiments.

In Table 7 and Table 8, we provide the augmentation results of FNOs using the learned symmetries, which are illustrated in Figure 6.

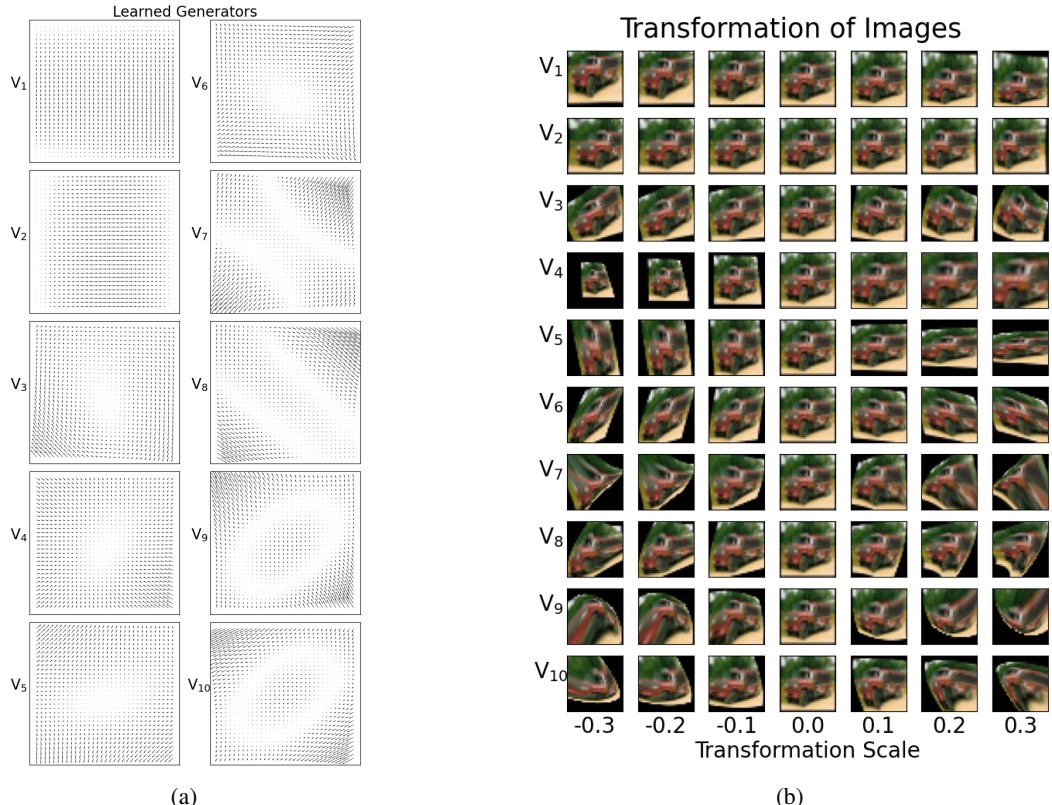

Figure 8: (a) The learned vector fields. (b) Transformed images using the learned generators. The images with transformation scale 0 are the original images.

Table 7: Comparison of augmentation methods with different numbers of data for KdV and KS

| | KdV | | | KS | | |
|---|---|---|---|---|---|---|
| # Data | None | Ground-truth | Ours | None | Ground-truth | Ours |
| $2^9$ | $0.398 \pm 0.013$ | $0.0640 \pm 0.0050$ | $0.0824 \pm 0.0052$ | $0.00693 \pm 0.00039$ | $0.00229 \pm 0.00014$ | $0.00614 \pm 0.00051$ |
| $2^7$ | $1.42 \pm 0.05$ | $0.246 \pm 0.012$ | $0.283 \pm 0.018$ | $0.324 \pm 0.031$ | $0.0241 \pm 0.0001$ | $0.0422 \pm 0.0002$ |
| $2^5$ | $4.47 \pm 0.26$ | $0.980 \pm 0.039$ | $1.12 \pm 0.10$ | $5.78 \pm 0.13$ | $1.14 \pm 0.10$ | $1.37 \pm 0.04$ |

# D    Technical Details

## D.1    Weighted Essentially Non-Oscillating (WENO) Scheme

This section is largely based on Zhang & Shu [37], Dumbser & Käser [12].

**WENO scheme in 1D.**    Consider a smooth function $f : \mathcal{X} \subseteq \mathbb{R} \to \mathbb{R}$, where we only have access to the function on the grid $\mathcal{X}_{\text{grid}} = \{x_1, \cdots, x_{N_{\text{grid}}}\}, x_1 < \cdots < x_{N_{\text{grid}}}$. To estimate the derivatives $f^{(k)}(x) = \frac{d^k f}{dx^k}(x)$ for some $x \in \mathcal{X}$, we use $k+1$ neighboring points in $I = \{x_i, \cdots, x_{i+k}\}$ and compute the unique $k$-th order polynomial $p_I(x)$ that interpolates the function $f$ on the points in $I$. In other words, we ensure $p_I(x_i) = f(x_i), \cdots, p_I(x_{i+k}) = f(x_{i+k})$. Then we can pick up the estimates of the derivatives using the polynomial as $f^{(k)}(x) \approx p_I^{(k)}(x)$. We call the set of neighboring points $I$ a *stencil*, and the polynomial $p_I$ the *reconstruction polynomial*. Usage of reconstruction polynomials is a fundamental concept in the numerical differentiation method known as the *finite difference method*.

In the weighted essentially non-oscillating (WENO) scheme, we compute multiple estimates $p_{I_1}^{(k)}(x), \cdots, p_{I_{N_s}}^{(k)}(x)$ for the derivatives $f^{(k)}(x)$ using multiple stencils $I_1, \cdots, I_{N_s}$. The estimates

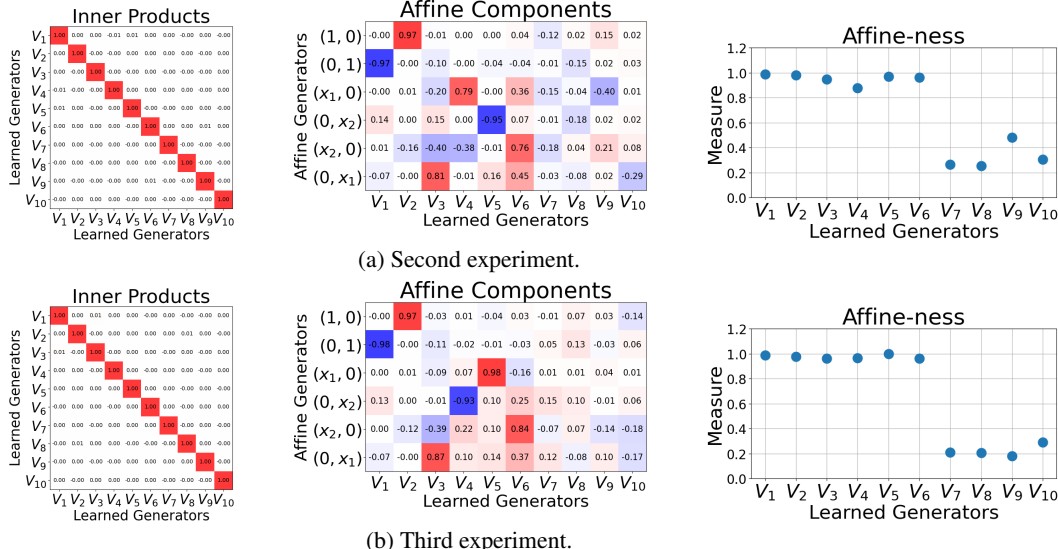

(a) Second experiment.

(b) Third experiment.

Figure 9: The same results (learned symmetries of images) under different initializations compared to Figure 4.

Table 8: Comparison of augmentation methods with different numbers of data for nKdV and cKdV

| # Data | nKdV | | cKdV | |
| | None | Ours | None | Ours |
|---|---|---|---|---|
| $2^9$ | $(3.52 \pm 0.11) \times 10^{-3}$ | $(2.58 \pm 0.17) \times 10^{-3}$ | $(3.72 \pm 0.33) \times 10^{-6}$ | $(3.44 \pm 0.37) \times 10^{-6}$ |
| $2^7$ | $(3.16 \pm 0.06) \times 10^{-2}$ | $(2.96 \pm 0.50) \times 10^{-2}$ | $(3.71 \pm 0.09) \times 10^{-5}$ | $(1.86 \pm 0.10) \times 10^{-5}$ |
| $2^5$ | $(4.79 \pm 0.09) \times 10^{-1}$ | $(2.44 \pm 0.11) \times 10^{-1}$ | $(7.90 \pm 1.21) \times 10^{-4}$ | $(2.99 \pm 0.37) \times 10^{-4}$ |

are then averaged with weights $\omega_1, \cdots, \omega_{N_s}$ which take account of the quality of each estimation. In the WENO scheme, we assume the grid points are sufficiently dense, hence the estimates are more accurate when the reconstruction polynomials $p_{I_m}$ are smooth. The smoothness of $p_{I_m}$ is measured by the *smoothness indicator*, defined as:

$$\text{IS}_{I_m} = \sum_{\alpha=1}^{k} \int_{\Delta} |\Delta|^{\alpha-1} (p_{I_m}^{(\alpha)}(x))^2 dx \tag{24}$$

where $\Delta = (x_j, x_{j+1}) \subset \mathbb{R}$ is an interval between two grid points containing $x$, and $|\Delta|$ is the length of $\Delta$. The weights $\omega_1, \cdots, \omega_{N_s}$ are defined as:

$$\omega_m = \frac{\alpha_m}{\sum_{m'} \alpha_{m'}}, \quad \alpha_m = \frac{\gamma_m}{(\epsilon + \text{IS}_{I_m})^b} \tag{25}$$

where $\gamma_m$ is called the *linear weight*, usually chosen heuristically, $b$ is a positive integer usually set to 2 or 4, and $\epsilon > 0$ is a small positive number preventing the denominator from being zero. The final estimate for the derivative $f^{(k)}(x)$ is computed by averaging the estimates $p_{I_1}^{(k)}(x), \cdots, p_{I_{N_s}}^{(k)}(x)$ with the weights $\omega_1, \cdots, \omega_{N_s}$:

$$\hat{f}^{(k)}(x) = \sum_{m=1}^{N_s} \omega_m \cdot p_{I_m}^{(k)}(x). \tag{26}$$

**WENO scheme for multivariate function.** The WENO scheme extends smoothly to multidimensional function $f : \mathbb{R}^n \to \mathbb{R}$. For $\boldsymbol{k} = (k_1, \cdots, k_n) \in \mathbb{Z}_{\geq 0}^n$, we denote $f^{(\boldsymbol{k})}(x) = \frac{\partial^k f}{\partial x_1^{k_1} \cdots \partial x_n^{k_n}}(x)$. On a stencil $I$ with an appropriate number of grid points, we compute the reconstruction polynomial $p_I(\boldsymbol{x})$ with a nonzero $\boldsymbol{k}$-degree term, and the smoothness indicator is defined similarly as:

$$\text{IS}_I = \sum_{1 \leq |\boldsymbol{\alpha}| \leq k} \int_{\Delta} |\Delta|^{|\boldsymbol{\alpha}|-1} (p_{I_m}^{(\boldsymbol{\alpha})}(\boldsymbol{x}))^2 d\boldsymbol{x} \tag{27}$$

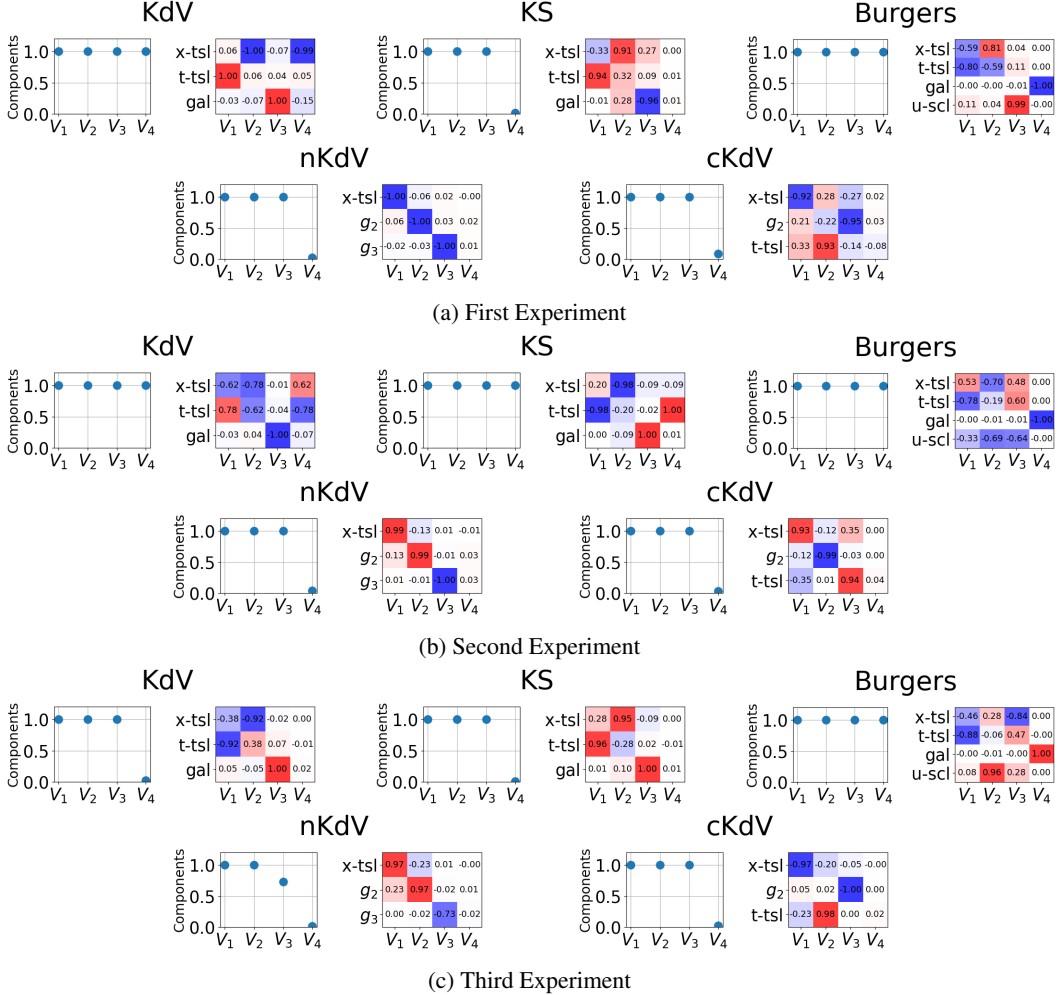

(a) First Experiment

(b) Second Experiment

(c) Third Experiment

Figure 10: Inner product comparison of the learned symmetry generators with the ground truth symmetries. (a), (b), and (c) are conducted with different random initializations.

where $\boldsymbol{\alpha} = (\alpha_1, \cdots, \alpha_n) \in \mathbb{Z}_{\geq 0}^n$, $|\boldsymbol{\alpha}| = \sum_{i=1}^n \alpha_i$ and the interval $\Delta$ is now a $n$-dimensional cell containing the point $\boldsymbol{x}$ with volume $|\Delta|$. Like the 1D case, the weights are defined using the linear weights and smoothness indicator.

**Choice of parameters.** We follow the choice of parameters of Dumbser & Käser [12]. The linear weight $\gamma_m$ is chosen to be 100 if $\boldsymbol{x}$ is inside the center cell of $I_m$ and 1 otherwise, reflecting the fact that the estimates are accurate as much as the point $\boldsymbol{x}$ is close to the center of $I_m$. The parameters $b$ and $\epsilon$ are chosen as $b = 4$ and $\epsilon = 10^{-6}$.

## D.2 Whittaker-Shannon Interpolation on a Periodic Domain

Let $\cdots, f[-1], f[0], f[1], \cdots$ be a discretization of a continuous signal on a real line. The Whittaker-Shannon interpolation recovers the original signal $f$ as:

$$f(t) = \sum_{n=-\infty}^{\infty} f[n]\mathrm{sinc}(t - n) \tag{28}$$

where $\mathrm{sinc}$ is the normalized sinc function $\mathrm{sinc}(t) = \frac{\sin(\pi x)}{\pi x}$. The *Nyquist-Shannon sampling theorem* states that states that $f(t)$ is the *perfect reconstruction* of $f$, in a sense that if the original function does not contain any frequencies higher than a certain threshold, called the *Nyquist frequency*, then $f$ is perfectly reconstructs the original signal.

Similarly, if $f[0], f[1], \cdots, f[N-1]$ is a discretization of a continuous signal on a periodic domain $[0, N]$ for some *even* positive integer $N$, the Whittaker-Shannon interpolation is given as:

$$f(t) = \sum_{n=-\infty}^{\infty} f[n \bmod N]\operatorname{sinc}(t-n) = \sum_{n=0}^{N-1} f[n] \sum_{n=-\infty}^{\infty} \operatorname{sinc}(t-mN) = \sum_{n=0}^{N-1} f[n]D_N(t) \quad (29)$$

where $D_N(t) = \sum_{n=-\infty}^{\infty} \operatorname{sinc}(t-mN)$ is the *Dirichlet kernel* with period $N$. The Dirichlet kernel $D_N(t)$ has a closed-form expression

$$D_N(t) = \frac{\sin(\pi t)}{N \tan(\pi t/N)}. \quad (30)$$

### D.3 Nonlinear Time Transformation of the KdV Equation

We apply a nonlinear time transformation $t$ defined as $t = t_0(e^{\frac{\hat{t}}{t_0}} - 1)$, where $t_0$ is a constant scaling factor, to the KdV equation $u_t + uu_x + u_{xxx}$. The derivative with respect to $t$ is transformed to

$$\frac{\partial}{\partial t} = \frac{\partial \hat{t}}{\partial t}\frac{\partial}{\partial \hat{t}} = e^{\frac{\hat{t}}{t_0}}\frac{\partial}{\partial \hat{t}} \quad (31)$$

hence the KdV equation with nonlinear time transformation (nKdV) becomes

$$e^{\frac{\hat{t}}{t_0}}u_{\hat{t}}u_t + uu_x + u_{xxx} = 0. \quad (32)$$

The three symmetries of the KdV changes accordingly. The space translation is left untouched, and the other two infinitesimal generators are tranformed as:

$$\frac{\partial}{\partial t} = e^{\frac{\hat{t}}{t_0}}\frac{\partial}{\partial \hat{t}} \quad (33)$$

$$t\frac{\partial}{\partial x} + \frac{\partial}{\partial u} = t_0(e^{\frac{\hat{t}}{t_0}} - 1)\frac{\partial}{\partial x} + \frac{\partial}{\partial u} \quad (34)$$

where we used the derivative notation $(1,0,0) = \frac{\partial}{\partial x}$, $(0,1,0) = \frac{\partial}{\partial t}$ and $(0,0,1) = \frac{\partial}{\partial u}$.

## E Ablation Studies

### E.1 Numerical differentiation.

When searching for symmetries of PDEs, we use the WENO scheme as a numerical differentiation method. We have found that the choice of numerical differentiator is critical, as gradients must flow through numerical differntiation during the backpropagation step. To compare different methods, we experimented with using the WENO scheme and another method: bilinear interpolation followed by discrete numerical differentiation on a regular rectangular grid.. When using bilinear interpolation followed by discrete differentiation, the loss failed to converge well, even with very small learning rate $10^{-6}$. The learned vector fields were not orthogonal, and they only spanned the time translation and the space translation at best, as in Figure 11. These experiments were conducted using the KS equation.

We hypothesize that the bilinear interpolation distributes each transformed point into two adjacent grid points, but discrete differentiation is done by repeatedly subtracting values of two adjacent grid points, so the gradient may not flow well during the backpropagation.

### E.2 Whittaker-Shannon interpolation.

In the PDE augmentation task, when the transformed PDEs are resampled into the regular rectangular grid, we use Whittaker-Shannon instead of more commonly used bilinear interpolation. This step is also crucial for training FNOs with transformed data, since FNOs are extremely sensitive to numerical error and hence any aliasing effects must be avoided. We compared the augmentation results using Whittaker-Shannon interpolation with those using bilinear interpolation, using the KS equation with 512 training data and report the result in Table 9. The results clearly demonstrate that bilinear interpolation adversely affects the training FNO models.

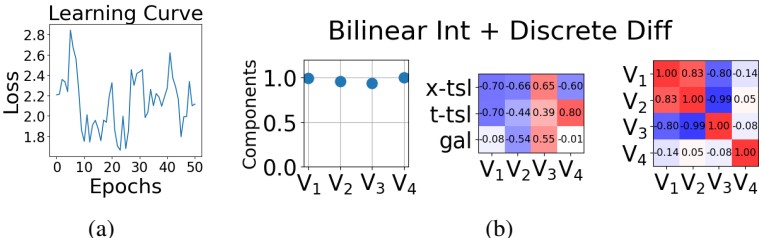

(a)                                        (b)

Figure 11: Experiment using using bilinear interpolation and discrete differentiation. (a) Learning curve. (b) Learned symmetries.

Table 9: Test NMSE comparison of augmentation using Whittaker-Shannon interpolation and bilinear interpolation.

|  | **No-aug** | **Whittaker-Shannon** | **Bilinear** |
|---|---|---|---|
| Test NMSE | $0.00693 \pm 0.00039$ | $0.006143 \pm 0.00051$ | $0.542 \pm 0.051$ |

# F   Hyperparameter analysis

In this section, we analyze the roles of various hyperparameters in our learning scheme. We use experiment setting of the KS equation in § 5.2. We selected the KS equation for analysis as it proved to be the most challenging in learning symmetries, possibly due to its fourth-order derivative term $u_{xxxx}$. In particular, among the three symmetries (space translation, time translation, Galilean boost) the algorithm easily learns the space and time translation in a few epochs, but learning the Galilean boost takes more epochs as in Figure 12. We evaluate the results by examining whether the three symmetries are correctly found in the first three slots.

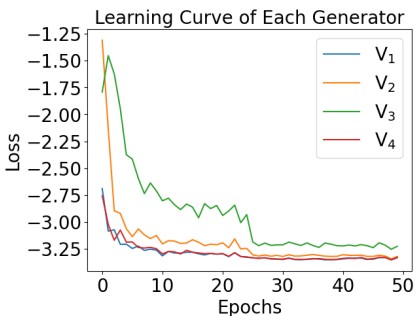

Figure 12: Learning curves of four symmetry generators.

## F.1   Lipschitz Threshold

The Lipschitz threshold $\tau$ in Equation 13 is the only parameter that constrains the function space in which we search for the symmetries. The threshold $\tau$ is set to $\tau = 3$ in the main experiments. We conduct additional experiments with $\tau = 1, 2, 4, 8, 16, 32$. Surprisingly, in all experiments, regardless of the value of $\tau$, we found the three ground truth symmetries.

## F.2   Transformation Scale

The transformation scale $\sigma$ controls how much we transform the data when searching for symmetries. The tuple $(x, t, u)$ in $\mathcal{X} \times \mathcal{U}$ is scaled so that $x$ and $t$ form a uniform grid on $[0, 1]^2$, and $u$ is scaled so that the standard deviation of $u$ closely matches that of $x$ and $t$, which is approximately $0.29$. Hence, the default transformation scale $\sigma = 0.4$ transforms data with slightly more than its standard deviation at most. We conduct experiments with various transformation scales $\sigma = 0.1, 0.2, 0.4, 0.6, 0.8$. The ground truth symmetries were found when $\sigma = 0.1, 0.2, 0.4$. When $\sigma = 0.6$, the Galilean boost was found but allocated in the fourth slot, meaning that the learning was unstable. When $\sigma = 0.8$, the model only found space and time translation correctly.

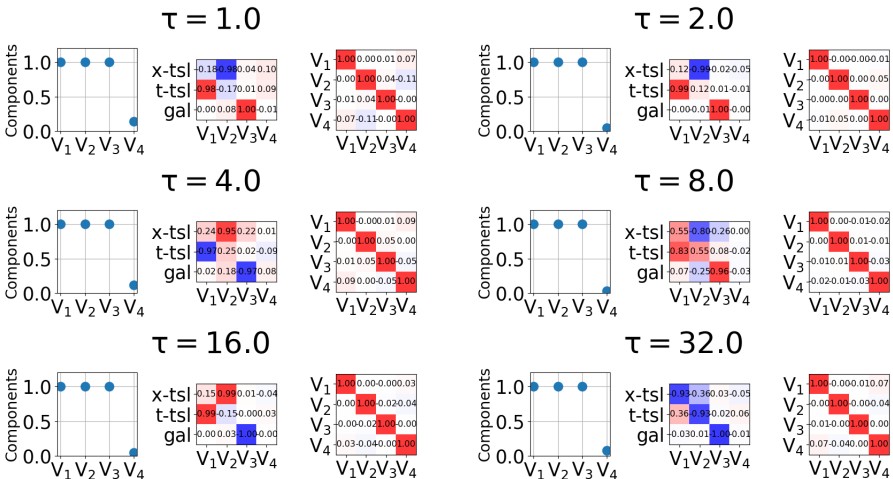

Figure 13: Experiments with various values of hyperparameter $\tau$.

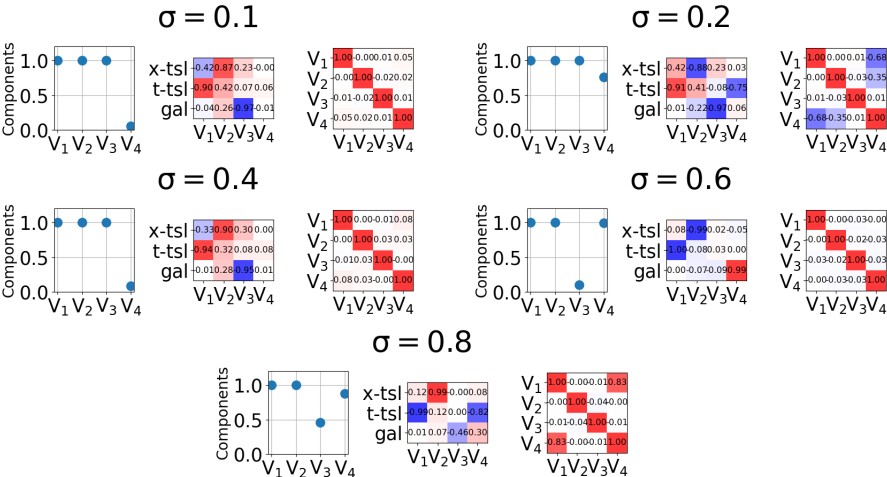

Figure 14: Experiments with various values of hyperparameter $\sigma$.

### F.3 Weights of Three Losses

Although the three loss terms Equation 14 are designed to have dimensionless scales, tuning the three loss terms in appropriate ranges is inevitable. We perform a grid search over the three weights $w_{\text{sym}}, w_{\text{ortho}}, w_{\text{Lipschitz}}$ of the total loss in a fixed grid $[1, 3, 10]^3$. We report the number of correctly learned symmetries across different values of $w_{\text{sym}}, w_{\text{Lips}},$ and $w_{\text{ortho}}$ in Table 10. The number 3 is the maximum number of symmetries to be discovered. We found that the ratio between $w_{\text{sym}}$ and $w_{\text{ortho}}$ significantly affects the results. In this case, the weight $w_{\text{ortho}}$ should be larger than $w_{\text{sym}}$ to prevent the learning of redundant infinitesimal generators.

## G Color-space Results

Recall that in our formulation, images take the form $\mathcal{X} \to \mathcal{Y}$ where $\mathcal{X} = [-1, 1]^2$ is the pixel space and $\mathcal{Y} \in \mathbb{R}^3$ is the RGB color space. Our formulation is not limited to searching for the symmetries on the 2D plane $\mathcal{X}$, but is capable of learning symmetries in the color space $\mathcal{Y}$. In this section, we elaborate how we modeled learning scheme for color-space symmetries and present the results.

Table 10: The number of correctly learned symmetries across values of $w_{\text{sym}}$, $w_{\text{Lips}}$, and $w_{\text{ortho}}$.

(a) $w_{\text{sym}} = 1$

| $w_{\text{ortho}}$ | $w_{\text{Lips}}$ | | |
|---|---|---|---|
| | 1 | 3 | 10 |
| 1 | 2 | **3** | **3** |
| 3 | 2 | **3** | **3** |
| 10 | **3** | **3** | **3** |

(b) $w_{\text{sym}} = 3$

| $w_{\text{ortho}}$ | $w_{\text{Lips}}$ | | |
|---|---|---|---|
| | 1 | 3 | 10 |
| 1 | 2 | 2 | 2 |
| 3 | **3** | **3** | **3** |
| 10 | **3** | **3** | **3** |

(c) $w_{\text{sym}} = 10$

| $w_{\text{ortho}}$ | $w_{\text{Lips}}$ | | |
|---|---|---|---|
| | 1 | 3 | 10 |
| 1 | 1 | 2 | 2 |
| 3 | 2 | 2 | 2 |
| 10 | **3** | **3** | **3** |

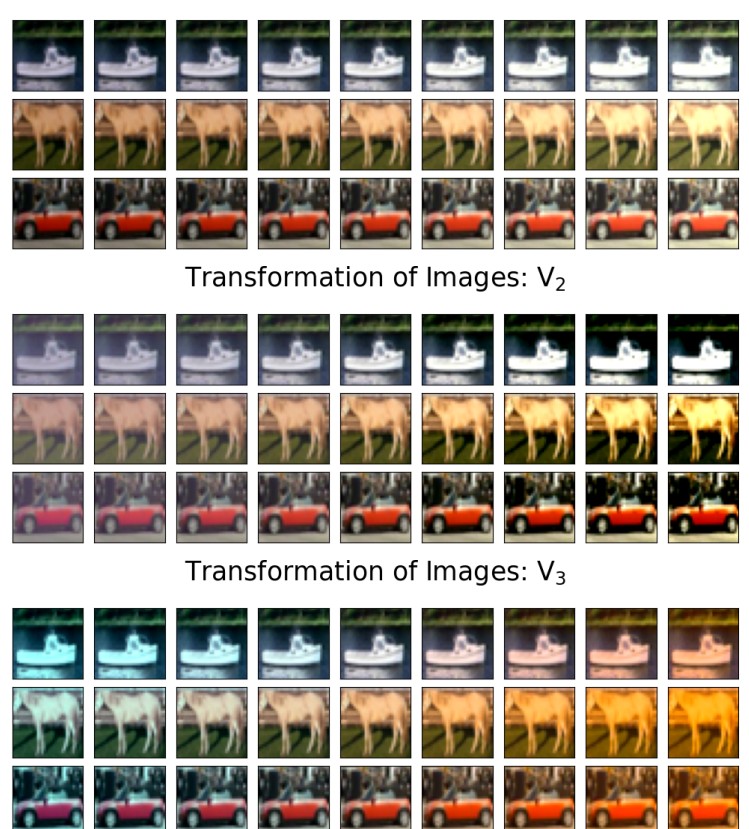

Transformation of Images: $V_1$

Transformation of Images: $V_2$

Transformation of Images: $V_3$

Figure 15: Learned color-space transformations.

### G.1 Learning Scheme

Similar to other cases, we model one-parameter groups on $\mathcal{Y}$ by a neural ODE, integrating over 3D vector fields $\mathcal{Y} \to \mathbb{R}^3$ parametrized by a neural network. Colors are non-uniform – for example, black, brown and gray appear much more frequently than pink or yellow in CIFAR-10 dataset. To address this, we first form a grid $\mathcal{Y}_{\text{grid}}$ on the color space $\mathcal{Y}$ by randomly sampling 1024 colors from the CIFAR-10 dataset. Then we define the weight function $w : \mathcal{Y} \to \mathbb{R}$, like the weight function in Equation 17, by the *color sensitivity* of the neural network $H_{\text{fext}}$. For a grid point $\boldsymbol{y}_i \in \mathcal{Y}_{\text{grid}}$, the weight $\omega(\boldsymbol{y}_i)$ is defined as:

$$\omega(\boldsymbol{y}_i) = \mathbb{E}_{f \sim \mathcal{D}} \left[ \left\| \frac{\partial H_{\text{fext}}(f)}{\partial \boldsymbol{y}_i} \right\| \right]. \tag{35}$$

To estimate the magnitude of the gradient with respect to the change of $\boldsymbol{y}_i$, we use neighboring points $\boldsymbol{y}_j \in \text{nbhd}(\boldsymbol{y}_i) \subset \mathcal{Y}_{\text{grid}}$. For a discretized image $\{\boldsymbol{x}_i, f(\boldsymbol{x}_i)\}_{x_i \in \mathcal{X}_{\text{grid}}}$, we use the nearest neighbor

algorithm to collect $f(\boldsymbol{x}_i) \in \text{Nearest}(\boldsymbol{y}_i)$ close to $\boldsymbol{y}_i$, and measure how much $H_{\text{fext}}$ changes when the color $f(\boldsymbol{x}_i)$ shifts towards the direction $\boldsymbol{y}_j - \boldsymbol{y}_i$:

$$\left\|\frac{\partial H_{\text{fext}}(f)}{\partial \boldsymbol{y}_i}\right\| = \frac{1}{|\text{nbhd}(\boldsymbol{y}_i)|} \sum_{\boldsymbol{y}_j \in \text{nbhd}(\boldsymbol{y}_i)} \left\|(\boldsymbol{y}_j - \boldsymbol{y}_i) \cdot \nabla_{\mathbb{1}_{\text{Nearest}(\boldsymbol{y}_i)}} H_{\text{fext}}(f)\right\| \tag{36}$$

where $\mathbb{1}_{\text{Nearest}(\boldsymbol{y}_i)}$ is a vector whose $i$-th entry is 1 if $f(\boldsymbol{x}_i) \in \text{Nearest}(\boldsymbol{y}_i)$ and 0 otherwise, and $\nabla_{\mathbb{1}_{\text{Nearest}(\boldsymbol{y}_i)}} H_{\text{fext}}(f)$ is computed by Jacobian-vector product.

Once the weight function is computed, we learn symmetries by optimizing through the validity score $S$, defined by cosine similarity of features of the learned neural network. Since there are (batch size) · (number of pixels) number of colors in a single batch of images, it's impractical to feed all of them directly into the MLP. Instead, we use the grid $\mathcal{Y}_{\text{grid}}$, and compute the vector field values on the grid and interpolate them using the nearest neighbor algorithm.

### G.2 Results

We open three slots for color-space symmetries and train the model with $w_{\text{sym}}, w_{\text{Lips}} = 1$ and $w_{\text{ortho}} = 3$. We found the brightness control, color contrast and blue-yellow shift respectively as in Figure 15.

