# OpenReview forum: "Learning Infinitesimal Generators of Continuous Symmetries from Data"
_NeurIPS.cc/2024/Conference — NeurIPS 2024 poster_

### Official Review · Reviewer_4Qv3 · 2024-06-14

**Soundness:** 3
**Presentation:** 3
**Contribution:** 2
**Rating:** 5
**Confidence:** 3

**Summary:**

This paper proposes using neural ODEs to parameterize symmetries by viewing the ODEs flow as an element of a one-parameter group. They show that by learning the parameters of the neural ODEs, they are able to recover ground truth symmetries in image classification and PDE tasks.

**Strengths:**

The paper is easy to read. The proposed ideas are clear, and appear to be mostly novel.

**Weaknesses:**

See questions below.

**Questions:**

1. Line 174: Do you also need "and for all $s\in[-\sigma,\sigma]$" in addition to the "for all $f\in\mathcal{D}$"?
2. Section 4.1: According to your definition of a symmetry $\vartheta$ (there exists $\sigma>0$ such that $\vartheta_s(f)$ is "valid" for all $f\in\mathcal{D}$ and all $s\in[-\sigma,\sigma]$), it seems like solving for $\vartheta^*$ from (4) does not guarantee that $\vartheta^*$ is actually a symmetry, even if the optimal value of (4) is less than $C$. That is, if the optimal value of (4) is less than $C$, then you only know that the validity score is less than the threshold on average, not for all data $f$ and all transformation scales $s$. Can you please clarify this aspect?
3. Line 217: Something is strange with the definition of $\vartheta_{\mathcal{X}}(f)$. In particular, what is $T_{\mathcal{X}}$, and why does $f$ not appear in the right-hand expression? It seems to me like perhaps you meant to define $\vartheta_{\mathcal{X}}(f)(x) = f(\vartheta_{\mathcal{X}}^{-1}(x))$, and hence that you need to assume the transformation $\vartheta_{\mathcal{X}}$ on $\mathcal{X}$ to be invertible. Can you please clarify whether this is a typo, or if I am missing something? Also, it is unclear what $\vartheta_{\mathcal{Y}}(f)$ is doing or why it is needed. Indeed, as written, you are defining two different transformations of $f$ in (8) (resulting in two different functions of $x$).
4. Line 226: How reasonable is it to assume that you know the actual number of possible symmetries for a given task? Is it possible for you to select $N_{\textup{sym}}$ to be too small, and then to miss out on learning some of the most important symmetries of a task?
5. Line 232: Typo ("orthonomral").
6. Line 230: In practice, are symmetries actually mutually orthogonal (as vector fields) to one another? It seems like this is not always the case per your Table 1, which shows that uniform scaling is not orthogonal to $x_1$-axis scaling. A comment on the validity of the orthogonality assumption underlying the use of this regularizer would be appreciated.
7. Line 243: Typo ("undersirable").
8. Line 246: Typo ("Lipshictz").
9. Line 280: "...the two translations are the most invariant one-parameter group..." Can you please explain in further detail how you come to this conclusion?
10. Table 2 and Figure 5: I suggest moving these to be centered with the rest of the text; it looks poorly placed in the right-hand margin as is, and causes Figure 5 values to be too small and hard to read.
11. Line 322: "We found the ground truth symmetries in all the experiments as in Figure 5..." It looks like you did not recover the true symmetries in the cKdV setting, as there are non-negligible components coming from each of the ground truth symmetries that appear in the learned symmetries.
12. Did you verify in your experiments that the learned "symmetries" are actually symmetries according to your definition using the validity score threshold? Overall, your experimental results in Figures 4, 5, and 10 seem to me like you are indeed learning orthogonal vector fields, and that sometimes the vector fields end up being equal to one of the ground truth vector fields, but other times not (turns out to be linear combination of multiple ground truths). This does not seem super convincing that you are reliably learning/recovering the ground truth symmetries.
13. Why did you introduce (8)? I don't see this transformation being used anywhere else in the paper.

**Limitations:**

N/A.

---

> ### Author Rebuttal · Authors · 2024-08-06
>
> Thank you for your feedback.
>
> **Q1. Necessity of phrase in line 174.**
> Yes. For all $f \in \mathcal{D}$ and for all $s \in [-\sigma, \sigma]$.
>
> **Q2. Clarification in Section 4.1.**
> To be more rigorous, our formulation leaves us a set of constraints $S(\vartheta^*_s,f) <C$ for all $f \in \mathcal{D}, s \in [-\sigma,\sigma]$, which is intractable to solve. Instead, with a mild assumption that non-symmetries have comparably higher values of the validity score than the true symmetries, we turn it into the proposed optimization problem. We will make this clearer in the revised version of our paper.
>
> **Q3. Typo in line 217.**
> Yes. This is a typo. it should be $(\vartheta_\mathcal{X}(f))(x) = f(\vartheta_\mathcal{X}^{-1}(x))$. We apologize for the critical typo and thank you for pointing it out.
>
> **Q4. The number of symmetries in line 226.**
> In our framework, we assume knowledge on a rough upper bound $N_{\text{sym}}$ on the true number $N_{\text{sym}}^*$. The stop-gradient operation in the orthonormality loss turns the optimization problem into a sequence of optimization problems with hierarchical constraints; the first slot is optimized without being affected by other slots, the second slot is only affected by the first slot, and so on. Consequently, the true symmetries are learned in the first $N_{\text{sym}}^*$ slots in the MLP. We believe having a rough estimate on an upper bound of the number of symmetries is reasonable.
>
> **Q6. Orthogonality assumption in line 230.**
> A set of symmetry generators form a vector space. For example, a sum of the unit $x_1$-axis translation generator and the unit $x_2$-axis translation generator gives a translation in the direction of the $45^\cdot$ vector. **Symmetry generators listed in Table 1 and Table 4 are in fact a basis of the space of symmetry generators.** Hence, if we set an inner product in the space of vector fields, there is always an orthonormal basis of symmetry generators.
>
> **Q9. Further details about the most invariant symmetries in line 280.**
> As discussed in the answer to Q4, the stop-gradient operation gives hierarchical constraints in the optimization problem, causing it to learn the earlier slots first and then the latter slots. Since the translations are found in the first and second slots, we conclude that they are “the most” invariant ones. This means that the outputs of the neural network are altered by the smallest magnitude along those symmetries.
>
> **Q11. True symmetries in cKdV in line 322.**
> As we elaborated in the answer to Q6, symmetry generators form a vector space. In the cKdV case, the exact and approximate symmetries form a vector space of dimension 3. We identified three orthonormal basis components of that vector space. Although these components do not match the closed-form expressions exactly, they span the same vector space as the closed-form symmetries. Therefore, we conclude that the three symmetries have been correctly learned.
>
> **Q12. Evidence for learning actual symmetries.**
> Again, symmetry generators form a vector space, so learning linear combinations can be as well regarded as learning the true symmetries. Moreover, we compare the distributions of validity scores and report them, both with images (top) and PDEs (bottom) in the tables below, where the non-symmetries are modeled by neural networks with random initialization. The values of quantile 0.95 shows that the true symmetries indeed have lower validity scores than non-symmetries.
>
> |  | mean | median | quantile 0.9 | quantile 0.95 |
> |---|---|---|---|---|
> | symmetry (x-translation) | 0.036 | 0.024 | 0.085 | 0.114 |
> | non-symmetry | 0.092 | 0.049 | 0.224 | 0.289 |
>
> |  | mean | median | quantile 0.9 | quantile 0.95 |
> |---|---|---|---|---|
> | symmetry (galilean boost) | 0.133 | 0.035 | 0.435 | 0.570 |
> | non-symmetry | 95.9 | 0.859 | 29.0 | 132.1 |
>
> **Q13. Necessity of equation (8).**
> The equation (8) is not explicitly used in implementation, but it states how transformations on the spaces $\mathcal{X}$ and $\mathcal{Y}$ induce a transformation of the function $f:\mathcal{X} \rightarrow \mathcal{Y}$. For example, an image can be viewed as a function $f:\mathbb{R}^2 \rightarrow \mathbb{R}^3$, which is then discretized as $ \\{ f(x\_i) \\} $ on pixels $\mathcal{X}\_{\text{grid}} = \\{ x_i \\}$. Transformation of pixels $x\_i \mapsto \vartheta_\mathcal{X}(x\_i)$ results in a transformed function $\tilde{f}:\vartheta\_{\mathcal{X}}(x\_i) \mapsto f(x\_i)$, which corresponds to the function described in Equation (8).

---

### Official Review · Reviewer_7EZa · 2024-07-08

**Soundness:** 2
**Presentation:** 3
**Contribution:** 3
**Rating:** 6
**Confidence:** 4

**Summary:**

The paper pertains to the topic of data-driven symmetry discovery. The authors propose a method allowing symmetry discovery beyond pre-defined Lie groups, by learning to transform datapoints, potentially in a non-affine manner, via a learned ODE (referred to as the *one-parameter group*, where the single parameter is the time variable in the ODE), the velocity field of which is typically parametrised by an MLP.

Crucially, to optimise this, the authors choose an objective - *validity score*, which is a predefined measure of the extent to which a transformed datapoint is symmetric to the input one (in their examples: for images, they use the cosine similarity between features extracted by a pretrained NN, while for PDEs, they measure the value – error – of the PDE for the transformed datapoint). Additional regularisers are used to ensure the diversity of the symmetries learned (orthogonality between different learned velocity fields) and smoothness (minimisation of an estimate of their local Lipschitz constants). Experimentally, the method is tested on image classification (CIFAR10) and PDE solving (KdV, KS, Burger’s equations) showing that known symmetries are retrieved along with additional approximate symmetries, while the learned symmetries are subsequently used for data augmentation showing competitive results to methods using pre-defined augmentations.

**Strengths:**

**Significance** . The paper studies an important problem (*data-driven symmetry discovery*) in machine learning, but also physical sciences where symmetries are abundant but potentially unknown. Identifying unknown symmetries and incorporating them in downstream ML models (e.g. via augmentation) can improve generalisation, especially in low-data regimes, while additionally, it can potentially provide novel insights about the task at hand.

**Novelty/Generality**
- The presented methodology has the capacity to recover symmetries arising from *non-affine* data transformations. This is contrary to prior work, where mostly linear/affine transformations are dealt with.
- Additionally, this method does not require making assumptions about the structure of the target group. This is common in prior art, where typically a subgroup of a predefined group is learnt.
- The authors take advantage of well-established concepts that are underexplored by the ML community (e.g. modelling transformations via the one-parameter group) - this helps to broaden the available toolbox in the field of ML & symmetries/ equivariant ML.

**Execution/Implementation**
- Although the proposed method has multiple complicated components (NeuralODEs, difficult objective to optimise for), it is nonetheless well-executed yielding competitive results and recovering known symmetries in popular testbeds.

**Weaknesses:**

**Applicability and scope**. Perhaps the biggest limitation of the proposed method is the *reliance on the validity score*. Although the authors claim to be able to learn symmetries by making as few assumptions as possible (see strengths), this seems to be contradicted by the need to manually design a validity score. Moreover, I have the impression that the validity score is not merely a hyperparameter, but it is decisive for the symmetries that will be learnt (basically it is the objective function of the optimisation problem).
-  For example, in the case of images, the choice seems ad hoc (closeness in the representation space of a pre-trained encoder NN). What leads the authors to believe that the features of equivalent (symmetric) images extracted from the pre-trained NN should be close? Have the authors tried to verify this assumption? I think the empirical validation is insufficient here (section 5.1.), so I am not entirely convinced.
- In general, I do not see a generic way to define validity scores and perhaps the authors have slightly overclaimed in that respect. I would like to read the authors' viewpoints on that. For PDEs, the validity scores are indeed reasonable and generic, so perhaps, they would like to put more emphasis on this perspective.

Furthermore, the authors introduce the concept of learning symmetries via the one-parameter group, claiming that it is more general than prior parametrisations that can only learn linear/affine groups. However, it is unclear what the present parameterisation can express, e.g. does it allow learning any continuous group or implicit assumptions are made here as well?
- Additionally, could the authors discuss if it would be possible to learn finite groups with this method as well and if not, how could those be incorporated?


**Related Work/Comparisons**. The work of Forestano et al., MLST’2023 is quite relevant to the present manuscript, with the main difference being that in that work, the transformations are parameterised by an MLP instead of a NeuralODE (the oracle used in this work seems similar to the validity score used here). Since the two works have many similarities, I think that the authors should discuss in more detail their differences and the advantages of their work (e.g. as far as I understand the MLP cannot guarantee that the transformations form a group). Note that modelling transformation via an MLP (or any NN in general) instead of a NeuralODE seems more straightforward and probably easier to train and more computationally friendly.

**Experiments**. I believe some additional empirical evidence would strengthen the authors' claims.
- Most importantly, an experimental comparison against the type of parameterisation used in Forestano et al. (MLP) should be provided, to verify if NeuralODEs are indeed a more appropriate parameterisation.
- Moreover, baselines are mostly missing, e.g. comparing against other methods for data-driven symmetry discovery (I am not super familiar with these works, but if I am not mistaken LieGAN by Yang et al., ICML’23 is a recent example).
- The reported results after augmenting with the learned symmetries do not seem to improve significantly compared to known/default augmentations. Can the authors discuss why this might be the case? This is important since it might undermine the necessity of the proposed approach.  To be more convincing, perhaps the authors should perform experiments on problems where the symmetries are not known a priori.
- Additionally, ablation studies seem to provide important insights but are only discussed in the appendix. I would recommend being more upfront in the main paper and discussing in the rebuttal the following: sensitivity to hyperparameters (multiple are needed: loss coefficients, $\sigma$ and $\tau$), the method for choosing them, the difficulty of optimisation  (3 terms are used in the objective) and if all losses are optimised in a balanced manner. Similarly, for the parameter $N_sym$, which is now chosen based on prior knowledge of the number of existing symmetries.

**Presentation/Exposition**. (disclaimer - this is a minor weakness) Given that the notions discussed here are probably not widely known across the ML community, I believe that the authors should aim to provide more in-depth explanations to make their work more accessible. For example,
- Multiple group theory/symmetry concepts are discussed without definitions (group generators, Lie group, Lie algebra, Lie bracket etc.). Additional examples that are not well-contextualised include in section 2 the one-parameter group discussion, the Lie algebra of the affine group and the discussion on the PDE symmetries (Lie point symmetries etc.). Adding some references here and providing some examples for the mentioned PDE symmetries would help.
- In section 5.2., some concepts regarding the experimental details are mentioned without appropriate explanations, while others are only mentioned in the appendix, although it appears that they are crucial for the method. Perhaps the authors should be more upfront and explanatory regarding the aforementioned.

**Questions:**

- How is the weight function in L234 defined, and how important is this for optimisation? Are different weight functions ablated?
- It’s unclear why the stop-gradient is needed in L237. I did not find the justification fully convincing. Could the authors elaborate? What happens experimentally if one does not use stop-gradient?
- Although intuitive, it’s unclear why the inline Eq. in L212 holds for negative $\alpha$.

**Suggestion**.
In case the authors do want to present their method as generic, I think a deeper experimental evaluation in data beyond PDEs would help a lot (e.g. testing on other image datasets, ablating different validity scores etc).

**Minor**:
- What if the chosen validity score is not differentiable?
- *Notation*.
  - The notation $\theta^V_s(x)$ is a bit dense (why use V as a superscript?). Eq (1) is a bit confusing. The inline Eq in line 83 sees clearer to me.
  - Notation in sec. 4 could be also simplified a bit or made more accessible with examples (e.g. 4.1: give an example for $\mathcal{A}$, i.e. the set of all natural images).

**Limitations:**

Some of the limitations are adequately discussed. An important missing point is, in my opinion, the need to manually design the validity score in domains beyond PDEs.

No foreseeable negative societal impact.

---

> ### Author Rebuttal · Authors · 2024-08-06
>
> Thank you for your feedback.
>
> **Weakness 1, 2. Reliance on the validity score.**
> We see the requirement of a validity score as a trade-off for not requiring the predefined set of symmetry generators, and searching for symmetries across the entire class of continuous transformations, which increases the degree of freedom of search space by an infinite amount. In general, a validity score is a quantification of some criterion that transformed data should meet. If one aims to find symmetries of a given dataset, they should first question “what do symmetries of the dataset truly mean”, and then should come up with a criterion that the symmetries should meet. Our validation score is a quantification of that criterion.
>
> For PDEs, the proposed validity score comes out quite naturally. For image data, while it might seem ad-hoc at a first glance, we argue that the proposed validity score is still based on the principle shared with PDE case – the invariance measure of the targets we wish to learn (PDE solutions for PDE data and features extracted from the encoder for image data). That is, the generic principle behind the validity score is to measure invariance of a target function with respect to transformations. For PDEs, we use the L1 error to measure this invariance. For images, we adopt the cosine similarity, which is a popular metric for measuring feature distances. We also note that other methods learning symmetries from images also have similar counterparts, such as the discriminator in LieGAN [2] and the classifier in Augerino [1] and LieGG [3].
>
> **Weakness 3. Range of the parametrization.**
> Our parametrization can model any continuous (or more rigorously, differentiable) groups acting on Euclidean space, given that neural networks, according to the universal approximation theorem, can approximate any continuous vector fields. The only additional inductive bias imposed on our model is Lipschitz continuity of the vector field to be learned. Below is a proof sketch of this argument. Note that we consider connected Lie groups, so discrete groups such as reflections are not considered.
>
> Let’s say a connected Lie group $G$ acts on a Euclidean domain $\mathbb{R}^n$ as $g:x \in \mathbb{R}^n \mapsto g\cdot x$, with corresponding Lie algebra $\mathfrak{g}$ and the exponential map $\exp:\mathfrak{g} \rightarrow G$. By the properties of connected Lie groups, any group element $g \in G$ can be expressed as $g = \exp(\epsilon_1\alpha_1) \cdots \exp(\epsilon_k\alpha_k)$ for $\alpha_1,\cdots,\alpha_k \in \mathfrak{g}$ and $\epsilon_1,\cdots,\epsilon_k \in \mathbb{R}$. Each component $\exp(\epsilon_i\alpha_i)$ for $i=1,\cdots,k$ can be seen as an action of a one-parameter group defined by $(t,x) \mapsto \exp(t\alpha_i)\cdot x$, with the infinitesimal generator given by $V_i(x) = \frac{d}{dt}|_{t=0} (\exp(t\alpha_i)\cdot x)$.
>
> **Weakness 4. Discrete groups.**
> Unfortunately, our method cannot learn finite groups. As in the image task, if there exists a known finite group symmetry, it can be incorporated as augmentation.
>
> **Related work/Comparisons and Experiments 1, 2. Baselines.**
> Please refer to paragraph “Comparison with baselines” in the general comment.
>
> **Experiment 3. Augmentation performances.**
> Since we parametrize the symmetry generators by MLPs, they may exhibit numerical instability. So the closed-form symmetries are expected to perform slightly better.
>
> **Question 1. importance of the weight function.**
> As described in Appendix A.1., the choice of weight function is crucial in the image task since each pixel has a different level of importance when put in a learned neural network. For example, if we use a constant weight $\omega \equiv 1$, then we may end up learning transformations that only move the pixels near the boundaries by a large amount. We compute a pixel sensitivity function using jacobian-vector products and use it as a weight function.
>
> We conducted additional experiments ablating different weight functions. We tested two handcrafted weight functions: the “plain weight” defined by $\omega(x) = 1$ for all $x\in [-1,1]^2$, and the “step weight” defined by $\omega(x) = 5$ if $||x||<0.25$, $\omega(x) = 4$ if $0.25<||x||<0.5$, and so on, decreasing in steps until $\omega(x) = 1$ if $||x||>1$. The weight functions and the results are depicted in Figure 3 and Figure 4 in the general comment. Indeed, the correct affine symmetries are not found with these weight functions.
>
> **Question 2. Role of stop-gradient operation.**
> In our framework, we assume knowledge on a rough upper bound $N_{\text{sym}}$ on the true number $N_{\text{sym}}^*$. The stop-gradient operation in the orthonormality loss turns the optimization problem into a sequence of optimization problems with hierarchical constraints; the first slot is optimized without being affected by other slots, the second slot is only affected by the first slot, and so on. Consequently, the true symmetries are learned in the first $N_{\text{sym}}^*$ slots in the MLP. Stop-gradient operation reduces dependence of our algorithm on the hyperparameter $N_{\text{sym}}$, and without the stop-gradient operation, we would learn mixtures of true symmetries and non-symmetries when $N_{\text{sym}}$ is larger than $N_{\text{sym}}^*$.
>
> **Question 3. One-param group in the negative direction.**
> Solutions of ODEs $\frac{d}{ds}\vartheta_s^V(x) = V(x)$ and $\frac{d}{ds}\vartheta_s^{-V}(x) = -V(x)$ with initial conditions $\vartheta_0^V(x) = \vartheta_0^{-V}(x) = x$ are related by $\vartheta_s^{V} \equiv \vartheta_{-s}^{-V}$ due to uniqueness of ODE solutions.
>
> **Presentation/Exposition.** Thank you for your feedback. We will incorporate your suggestions in the revised version of our paper.

---

> > ### Comment · Reviewer_7EZa · 2024-08-12
> > **Post rebuttal**
> >
> > I thank the authors for their response.
> >
> > - Regarding the validity score, unfortunately, I did not find the response satisfactory. Arguably, choosing the validity score is a central element of this work, therefore, I strongly encourage the authors to be upfront about it and clarify that this is a limitation of the approach and that (perhaps strong) assumptions are needed in domains beyond PDEs, where the choice is natural.
> > - Regarding the comparison with baselines, I was not able to locate an exact response to my concern. Simply put, my point was to replace the NeuralODE with an MLP (no ODE integrator, just direct prediction of the transformation). This is a natural baseline and would empirically support the necessity of employing a NeuralODE. Have the authors performed this experiment?
> > - I also encourage the authors to include in an updated version of their manuscript the discussion regarding the groups that can be learned (continuous and differentiable, but not finite).
> >
> > I keep my recommendation for acceptance, but as per my review, I have certain important reservations.

---

> ### Author Response · Authors · 2024-08-12
>
> Thank you for your comment.
>
> - Q1. We indeed acknowledge that choosing the validity score is critical and not direct. However, we clarify our stance: "symmetry" is not a fixed mathematical concept but a task-dependent jargon. For example, isometries of a metric space are transformations that preserve the metric, while symmetries of a physical system preserve the Hamiltonian or governing equations. Image symmetries are typically not rigorously defined but refer to transformations that maintain invariance to human perception; we replace "human eyes" with "neural network" in this context.
>
> - Q2. **The two experiments we have done are exactly that.** We replaced the neural ODE with an MLP that does not involve ODE integration, which now requires taking Lie derivatives (i.e. derivative along the vector field modeled by an MLP) to the validity score. We compared these setups using both image data and a toy experiment searching for isometries in Minkowski space. While using Lie derivatives (i.e., a plain MLP) fails with the image task, our method succeeds in both scenarios.

---

> > ### Comment · Reviewer_7EZa · 2024-08-14
> > **Recommendation**
> >
> > Thank you for your follow-up comments. Since many of my concerns have been addressed, I have increased my rating by 1 point.

---

### Official Review · Reviewer_UU8r · 2024-07-14

**Soundness:** 2
**Presentation:** 2
**Contribution:** 3
**Rating:** 6
**Confidence:** 3

**Summary:**

This paper proposes a symmetry learning algorithm based on transformations defined via infinitesimal generators. Using Neural ODE, an infinitesimal generator is learned that is capable of producing a sequence of transformed data through ODE integration. Validity score has been defined to check if the transformed data is valid wrt to a given task. For images, the validity score is picked to be cosine similarity while for PDEs the validity score is defined as the numerical errors in the original equations after the transformation. In addition to symmetry loss, two regularizations, orthonormality loss, and Lipschitz loss have been added, to remove trivial solutions. The authors present experiments on CIFAR10 and KdV equation and Burgers' equation in 1D for PDE.

**Strengths:**

1. The paper motivates the need for learning continuous symmetries well.
2. The idea presented in the paper is very neat and shows potential beyond the presented experiments.
3. This approach can learn both affine and non-affine symmetries in image classification and PDE tasks as shown in the experiments section.

**Weaknesses:**

The discussion on compute and model parameters comparisons with baseline missing. No other method was shown as a baseline in either of the experiments. There is some ambiguity in how exactly the validity score is used and in some cases, can be defined if the given task is equivariant.

**Questions:**

1. How does the validity score take into consideration, invariant tasks vs equivariant tasks?
2. Using cosine similarity between the extracted features for validity score, can be a problem for extreme transformations (as cosine values do not linearly change with parameter). How does this affect the learning of symmetry build on validity scores?
3. In Table 2, it is unclear how the validity score players out in learning the symmetries. Especially for default augmentation case. Could you please elaborate on this?
4. Total loss and loss-scale section: How is the true scale learned, if the inner product is normalized? Additionally if all the terms are affected by a data-dependent scale, shouldn't $w_{Lips}$ affect the overall weights?

### Clarifications
1. The figures can be made a little bigger.
2.  In line 16, 'whether the transformed data are still valid for the given task'. This phrasing is confusing.
3. In line 172, what does valid mean? simply put; does it comprise invariant transformations and approximately equivariant up to the threshold C?
4. Intuition on how to locate transformed data on grid is missing.

**Limitations:**

1. The authors do not compare with other methods of learning symmetry, as those methods use datasets that have explicit symmetry (like Rot-MNIST). Similarly for that of the PDE experiments. It would have been useful to have them in the paper.
2. The number of parameters and compute comparison across the proposed method and baseline approach like [9,14] is missing.  Also discussion the scale of numerical computation with an increase in dimension missing.

---

> ### Author Rebuttal · Authors · 2024-08-06
>
> Thank you for your feedback.
>
> **Question 1. Validity score in equivariant task.**
> We take a two-step approach: first, we learn symmetries based on validity scores, and second, we use these learned symmetries as augmentation and solve machine learning tasks. The validity score is designed to measure invariance of a certain function – the neural network in the image task, and the PDE residual values in the PDE task. We do not take equivariant tasks into account in this paper. However, if one were to design a validity score using equivariant neural networks, it could be achieved by modeling the transformation in both the input and output spaces and establishing an invariance criterion that corresponds to that equivariance.
>
> **Question 2. Cosine similarity and instability.**
> Cosine similarity values lie in $[0,1]$, and if arccos is applied, the angle values lie in $[0,\pi/2]$. The arccos function has an infinite gradient at 1, so the cosine similarity values must be clipped to a maximum $1-\epsilon$ for some small $\epsilon$. Except for this, we did not experience any instabilities coming from cosine similarity.
>
> **Question 3. Validity score and augmentation.**
> The validity score is only used when learning symmetries. Once the symmetries are found, we use the learned symmetries to augment the dataset for the target task – image classification task in this case. Table 2 shows comparison of augmentation performance with other commonly used augmentation methods.
>
> **Question 4. Scale of losses.**
> Sorry, but we didn’t fully understand your question. We will try to answer based on our understanding, but if you find it insufficient, please let us know.. The phrase “normalized inner product” in the paper refers to the following process: two vectors $v_1,v_2$ are normalized as $\hat{v_i} = \frac{v_i}{||v_i||}$ for $i=1,2$, and then the inner product $\langle \hat{v_1},\hat{v_2}\rangle$ is computed. Then we put arccos to make it lie in $[0,\pi/2]$. Similar process is applied for all terms, to make sure all losses are on a similar scale.
>
> **Clarifications 2, 3. Meaning of validity.**
> We say transformed data are valid for the given task when they are beneficial to the learning task. For example, in an image classification task, images rotated by small angles may help the task, but transformed images that are distorted too much may harm training. To approximate this validity, we design a function known as the validity score within our framework.
>
> **Clarification 4. Interpolation method.**
> We use bilinear interpolation for images, and Whittaker-Shannon interpolation for PDEs. Due to page limits, discussion on that is in Appendix A. and Appendix E.2., including discussion on why bilinear interpolation does not work for PDE data but Whittaker-Shannon works.
>
> **Limitations 1. Baselines.**
> Please refer to paragraph “Comparison with baselines” in the general comment.
>
> **Limitations 2. Computational costs.**
> We use a fairly small MLP, with around 150k parameters. Direct comparison of computational complexity with other methods is not feasible since all symmetry learning approaches have varying assumptions and goals. For example, L-conv [9] learns rotation by comparing rotated 7x7 random pixel images with the original ones.
>
> We provide some analysis on computational costs of our algorithm. The major computational burden of our algorithm arises from the ODE integration. Since we use the adjoint method for backpropagating [6], although the time complexity is $O(N_{\text{step}})$ where $N_{\text{step}}$ is the number of ODE steps, the memory requirement is $O(1)$. The choice of validity score also affects the complexity. However, the effect of dimension $n$ of the space is negligible, since it would only require changing the input and output dimension of the MLP by $\mathbb{R}^n \mapsto \mathbb{R}^n$.
>
> [4] Dehmamy et al., “Automatic symmetry discovery with lie algebra convolutional network.” NeurIPS 2021.
>
> [6] Chen et al., “Neural ordinary differential equations.” NeurIPS 2018.

---

### Author Rebuttal · Authors · 2024-08-06

We thank the reviewers for their insightful comments and suggestions. We are pleased that all the reviewers agree on the importance of symmetry discovery and find our research novel. Below, we address some commonly raised questions and present additional experimental results.

# Comparison with baselines
**(refer to performance comparison in the attached PDF file)**

There have been several attempts at symmetry discovery, each with different experimental settings and goals. The differences mainly arise from (i) what they aim to learn (e.g., transformation scales or subgroups from a larger group) and (ii) their assumptions about prior knowledge (e.g., complete, partial, or no knowledge of symmetry generators). Another important distinction is the viewpoint on symmetry: some methods learn symmetries that raw datasets inherently possess (implicit), while others learn symmetries from datasets explicitly designed to carry such symmetries (explicit).

Some recent symmetry discovery works are listed below. We emphasize that our method excels in two key aspects: **(i) our learning method reduces infinitely many degrees of freedom, (ii) our method works with high-dimensional real-world datasets.** For example, while LieGAN [2] and LieGG [3] reduce a 6-dim space (affine) to a 3-dim space (translation and rotation), ours reduces an $\infty$-dim space to a finite one. L-conv [4] also does not assume any prior knowledge, but it is limited in finding rotation in a toy task, where it learns rotation angles of rotated images by comparing them with the original ones, which are 7x7 random pixel images.

|  |Augerino [1]|lieGAN [2]|lieGG [3]|L-conv [4]|Forestano et al. [5]|Ours|
|-|-|-|-|-|-|-|
|symmetry generators|completely known|partially known (affine)|partially known (affine)|completely unknown|completely unknown|completely unknown|
|learn what?|transformation scale (e.g. rotation angle)|symmetry generator (rotation)|symmetry generator (rotation)|symmetry generator (rotation)|symmetry generator (in various low-dim settings)|symmetry generator (affine)|
|verified with what?|raw CIFAR-10|rotation MNIST|rotation MNIST|random 7x7 pixel images|toy data (dim <=10)|raw CIFAR-10|
|implicit or explicit|implicit|explicit|explicit|explicit|explicit|implicit|
|how?|optimize while training downstream task|compare fake/true data in GAN framework|extracts from learned NN using Lie derivative|compare rotated and original images|extracts from invariant oracle using Lie derivative|extracts from validity score using ODE integration|

Comparing those research with ours (i.e. without requiring any candidate symmetries, finding implicit symmetries, and using high-dimensional real-world datasets) requires highly nontrivial modification of those methods. For example, LieGAN learns six linear coefficients corresponding to six affine generators, which cannot be extended to ours. Thus, we compare our method with Forestano et al. [5] by learning symmetries in both a high-dimensional image dataset and a low-dimensional dataset using the Lie derivative and ODE integration. The experiment with the image dataset can also be seen as a comparison with LieGG.

**Ablation studies: learning symmetries of images using Lie derivative (comparison with LieGG and Forestano et al.)**

LieGG method extracts symmetries of images from a learned neural network using the Lie derivative and retrieves rotation from the rotation MNIST. The original LieGG assumes affine symmetry; however, for comparison purposes, we have adapted LieGG to our setting by using the Lie derivative instead of ODE integration in our method. As shown in Figure 1, this approach fails to learn the correct symmetries. Note that, to evaluate whether the learned symmetries $\\{V_1,\cdots,V_{10} \\}$ recover the space of true symmetries with an orthonormal basis $\\{L_1,\cdots,L_6\\}$, we follow this process: we first check whether $V_1,\cdots,V_6$ are mutually orthogonal. We then compute the values $\sqrt{\sum_{j=1}^6 \langle V_i, L_j \rangle^2}$ and check whether the first six such values are close to 1.

We describe our intuition on why the Lie derivative fails. The Lie derivatives are taken in the space of images, which are regarded as flattened RGB vectors. We claim that due to the high-dimensional nature of the image data, derivatives are less informative and much noisier than ODE integration. For example, rotating images moves the pixels, requiring the RGB values on the regular grid to be resampled by interpolating neighboring RGB vectors, such as with bilinear interpolation. Hence rotation of images in the space of flattened RGB vectors is a highly complicated and fluctuating non-smooth operation with noisy derivatives.

**Ablation studies: learning isometries of (3+1)-dimension Minkowski space using Lie derivative and ODE integration (comparison with Forestano et al.)**

We further demonstrate that usage of ODE integration is indeed capable of learning symmetries in the setting of Forestano et al.. We learn isometries of (3+1)-dimensional flat Minkowski space by setting the Lorentz metric $t^2 - x^2 - y^2 - z^2$ as a validity score and successfully find the six symmetries (three Lorentz boosts and three rotations), both using Lie derivative and ODE integration, as shown in Figure 2. This experiment shows that our approach works in both low-dimensional and high-dimensional settings, **while using Lie derivatives fails with high-dimensional image dataset.**

[1] Benton et al., “Learning invariances in neural networks from training data.” NeurIPS 2022.

[2] Yang et al., “Generative adversarial symmetry discovery.” ICML 2023.

[3] Moskalev et al., “Liegg: Studying learned lie group generators.” NeurIPS 2022.

[4] Dehmamy et al.,” Automatic symmetry discovery with lie algebra convolutional network.” NeurIPS 2021.

[5] Forestano et al., “Deep learning symmetries and their lie groups, algebras, and subalgebras from first principles.” MLST 2023.

---

### Decision · Program_Chairs · 2024-09-25

**Decision:**

Accept (poster)

**Comment:**

This paper uses neural ODEs (NODEs) to parameterize the infinitesimal generator of continuous symmetry groups.  The NODE is fit by defining and then optimizing a “validity score” that determines if the generated data is valid for the target application.  Experiments are reported on learning affine transformations of images and time translation, space translation, and Galilean boost symmetries for PDEs.  The reviewers had a generally favorable opinion of the work (2 x Weak Accept, 1 x Borderline Accept), with the reviewers appreciating that this is a general technique that can discover non-affine symmetries, whereas most previous work has either been constrained to affine or makes otherwise comparatively stronger assumptions about the group structure.  The two primary criticisms of the reviewers were that (1) the technique relies on defining a validity score, which may be unknown or hard to devise for some applications, and (2) the lack of comparison to baselines in the experiments.  My own assessment of these critiques is that, for the former, using a validity score is the natural price to pay for the lack of assumptions placed on the group and the symmetry type.  In other words, the structure has to come from somewhere, and I think the validity score is a reasonable way to do that.  Regarding the latter, in the rebuttal, the authors have provided a table that very nicely summarizes the relationship to prior work (and justifies the lack of empirical comparisons).  This has convinced me that the lack of baselines is reasonable.  The authors should include this table in the camera-ready version of the paper.